# Autoinverse: Uncertainty Aware Inversion of Neural Networks

**Navid Ansari**
Max Planck Institute for Informatics
Saarbrücken, Germany
nansari@mpi-inf.mpg.de

**Hans-Peter Seidel**
Max Planck Institute for Informatics
Saarbrücken, Germany
hpseidel@mpi-sb.mpg.de

**Nima Vahidi Ferdowsi**
Max Planck Institute for Informatics
Saarbrücken, Germany
nvahidi@mpi-inf.mpg.de

**Vahid Babaei**
Max Planck Institute for Informatics
Saarbrücken, Germany
vbabaei@mpi-inf.mpg.de

## Abstract

Neural networks are powerful surrogates for numerous forward processes. The inversion of such surrogates is extremely valuable in science and engineering. The most important property of a successful neural inverse method is the performance of its solutions when deployed in the real world, i.e., on the *native forward process* (and not only the learned surrogate). We propose `Autoinverse`, a highly automated approach for inverting neural network surrogates. Our main insight is to seek inverse solutions in the vicinity of reliable data which have been sampled form the forward process and used for training the surrogate model. `Autoinverse` finds such solutions by taking into account the predictive uncertainty of the surrogate and minimizing it during the inversion. Apart from high accuracy, `Autoinverse` enforces the feasibility of solutions, comes with embedded regularization, and is initialization free. We verify our proposed method through addressing a set of real-world problems in control, fabrication, and design. Our code and data are available at: https://gitlab.mpi-klsb.mpg.de/nansari/autoinverse

## 1 Introduction

"... optimizing for the wrong thing — offering prayers to the bronze snake of data rather the larger force behind it." [6]

With the deep learning breakthrough during the last decade, data-driven modeling using neural network based *surrogates* is now a standard practice in science and engineering. These surrogates strive to imitate the behavior of a *native forward process* (NFP), such as a physics simulation, which maps a *design* into its *performance*[1]. While forward processes are essential for troubleshooting and analysis, oftentimes their ultimate application lies in their inversion, i.e., the reverse process of mapping performances into designs. Despite the recent progress in inversion of neural networks within multiple disciplines [35, 16, 13, 37], a fundamental unaddressed question is still standing out. Inversion of a surrogate model, fitted to the data sampled from the NFP, is ultimately different than the inversion of the NFP itself. The source of this gap could be explicit, such as the noise in

---

[1]In the applications showcased in this paper (fabrication-oriented design and robotics), the term *design* refers to the space where the input to the forward process is characterized and parameterized and *performance* refers to the parameterized space of desired properties. Commonly, *hidden state* or *parameters*, and *measurement* or *goal* are used interchangeably with design and performance, respectively

36th Conference on Neural Information Processing Systems (NeurIPS 2022).

measurements, or implicit, such as the poor sampling of the NFP. Although the obtained solutions from inverting the surrogate can be evaluated on the NFP, none of the current neural inversion methods offers a tailored solution for addressing this important gap.

Our main insight in this work is to *expect* and *account* for any potential mismatch between the data, and consequently the surrogate, on the one hand and the NFP on the other. Our proposed method, `Autoinverse`, realizes this vision by taking into account the predictive uncertainty of the surrogate and minimizing it during the inversion. Therefore, the inverted solutions avoid the unreliable regions within the training data.

We show that our `Autoinverse` strategy can augment existing neural inversion methods (both optimization-based and architecture-based approaches) with uncertainty compensation in a simple and practical manner. `Autoinverse` closes the gap between the surrogate and the NFP *not* through attempting a perfect fit of the surrogate to the NFP, an onerous task, but by finding inverse solutions in the vicinity of the reliable training data where the surrogate and the NFP are most similar. Neural inverse methods equipped with `Autoinverse` outperform their counterparts significantly on both standard data sampled from the NFP and imperfect data, e.g., those corrupted by noise. Apart from high accuracy, `Autoinverse` enforces the feasibility of solutions, comes with embedded regularization (freeing the inversion approaches from hand-crafted regularizations based on domain knowledge), and is initialization-free. It achieves all these properties in a highly automated manner and only with a light, intuitive tuning.

## 2 Related work

**Neural Network Inversion**    We can divide neural network inversion approaches into two main categories. First, *inverse architectures*, where we compute a network architecture that takes a given performance and maps it into a (distribution of) design(s). Second, *direct optimization* [33], where we optimize for a design such that it produces the desired performance. Although the simplest inverse architecture can be attempted by training a neural network in the reverse direction, it fails because of the one-to-many nature of the mapping. *Tandem* networks use an inverse architecture by employing a pre-trained forward network in order to compute a consistent loss. The tandem approach has been developed independently across different disciplines [23, 32, 33, 39] dating back (at least) to Tominaga [34]. Many inverse architectures try to model the conditional posterior, $p(x|y)$, using variational methods [24, 19] based on (conditional) variational auto-encoders [20]. These networks condition the design on the target performance and yield a distribution of solutions from which multiple samples could be drawn. Kruse et al. [21] show that the invertible neural networks (INNs), built upon normalizing flows [10], give the highest accuracy in terms of both surrogate error and design posterior compared to a wide range of inverse architectures.

When using direct optimization methods, gradient-based optimizers can be readily used as the neural surrogate is differentiable. Ren et al. [30] use stochastic gradient descent via backpropagation with respect to the design variables, and present a highly accurate and practical method. They benchmark their method, dubbed as *neural adjoint* (*NA*), against a set of inverse architectures and obtain significantly more accurate solutions. Sun et al. [33] showed a similar approach except using a quasi-Newton method for optimization. Ansari et al. [2] push forward in this direction by demonstrating that, for piecewise linear neural networks, e.g., those with ReLU activation, the direct optimization can be formulated as a mixed-integer linear program (MILP) and thus obtain *globally* optimal solutions. While optimization methods are very accurate, their main disadvantage is their performance as they can be orders of magnitude slower than inverse architectures.

**Neural Networks and Predictive Uncertainty**    While neural networks are ubiquitous in almost all branches of natural sciences, their weakness at quantifying predictive uncertainty impedes their use in crucial applications. Using a Bayesian formalism [4], Bayesian neural networks (BNNs) [28, 9, 26], given the training data and a prior over network's parameters, compute the posterior distribution of the parameters. Having computed the posterior, the predictive uncertainty can be computed. The inference step in computing BNNs is known to be computationally hard [17]. This explains the popularity of simpler methods for estimating predictive uncertainty, such as Monte Carlo dropout [11] and Deep Ensembles [22]. Deep Ensembles strikes a good balance between simplicity and practicality on the one hand and predictive performance on the other hand (Section 3.3). One of the main advantages of the Deep Ensembles is its capability to predict aleatoric and epistemic uncertainty separately.

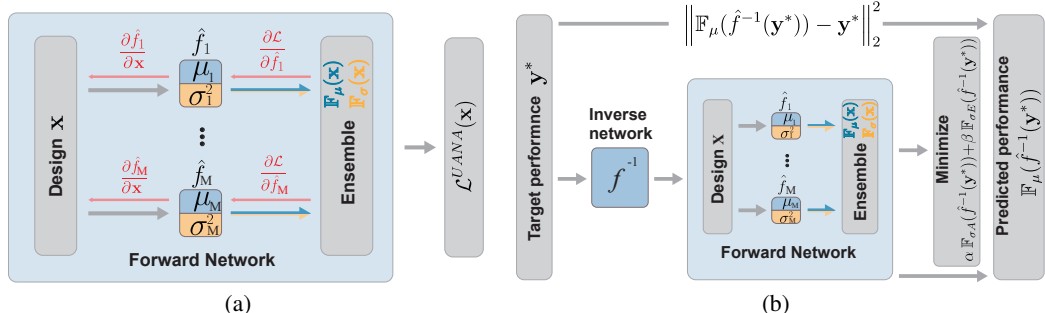

Figure 1: By using a deep ensemble predictive uncertainty estimator as the forward model we can make conventional inversion methods uncertainty aware. On the left we can see `UANA` and on the right `UA-tandem` architecture.

Aleatoric and epistemic uncertainty carry different information regarding the surrogate [18] and considering both of them improves the quality of the neural inversion (Section 4.3). Neural networks capable of predictive uncertainty are increasingly adopted in many different applications, such as reinforcement learning and active learning [12]. As we shall see, neural inversion is yet another domain that takes advantage of this trend.

## 3 Method

`Autoinverse` is an easy-to-implement technique for augmenting neural inverse methods with uncertainty awareness. `Autoinverse` achieves this goal by, first, training a surrogate capable of predictive uncertainty [25]. Second, relying on this trained surrogate and using a novel inversion cost function, `Autoinverse` finds accurate designs with minimal uncertainty. We apply `Autoinverse` on two inverse methods belonging to the two main neural inversion categories, i.e., optimization- and architecture-based in Sections 3.1 and 3.2, respectively. As we will see in Section 3.3, we rely on established methods [22] to train networks equipped with predictive uncertainty.

### 3.1 Uncertainty aware neural adjoint (`UANA`)

Given a pretrained neural surrogate $\hat{f}(\cdot)$, neural adjoint (`NA`) [30] is an inverse method that uses the cost function $\mathcal{L}^{NA}(\cdot)$ to push designs $\mathbf{x}$ to have a performance $\hat{f}(\mathbf{x})$ as close as possible to a desired performance $\mathbf{y}^*$:

$$\mathcal{L}^{NA}(\mathbf{x}) := \arg\min_{\mathbf{x}} \left\| \hat{f}(\mathbf{x}) - \mathbf{y}^* \right\|_2^2. \tag{1}$$

`NA` uses gradient descent to iteratively reduce the cost function, a scheme much like training neural networks but with the input as the optimization variable instead of network's weights and biases. Equation 2 shows a single `NA` iteration with $\delta$ as the step size:

$$\mathbf{x}^z = \mathbf{x}^{z-1} - \delta\left(\frac{\partial \mathcal{L}^{NA}}{\partial \hat{f}} \times \frac{\partial \hat{f}}{\partial \mathbf{x}}\right). \tag{2}$$

`Autoinverse` proposes to perform the inversion using a pretrained BNN. We use Deep Ensembles [22] made of $M$ neural networks capable of a prediction $\mathbb{F}_\mu(\mathbf{x})$, as well as its *aleatoric* $\mathbb{F}_{\sigma A}(\mathbf{x})$ and *epistemic* $\mathbb{F}_{\sigma E}(\mathbf{x})$ uncertainties (Section 3.3). Aleatoric uncertainty increases as the noise level in the training data increases. Epistemic uncertainty measures the uncertainty in the model. `Autoinverse` modifies `NA` such that we obtain solutions $\mathbf{x}$ that have performances close to the target performance $\mathbf{y}^*$ *while* resulting in small aleatoric and epistemic uncertainties:

$$\mathcal{L}^{UANA}(\mathbf{x}) := \arg\min_{\mathbf{x}} \|\mathbb{F}_\mu(\mathbf{x}) - \mathbf{y}^*\|_2^2 + \alpha\,\mathbb{F}_{\sigma A}(\mathbf{x}) + \beta\,\mathbb{F}_{\sigma E}(\mathbf{x}) \tag{3}$$

We introduce $\alpha$ and $\beta$ as hyperparameters to adjust the relative significance of aleatoric and epistemic uncertainties, respectively.

Equation 4 shows how one iteration of `UANA` requires the back-propagation using the ensemble of all gradients of $M$ networks of Deep Ensembles:

$$\mathbf{x}^z = \mathbf{x}^{z-1} - \delta \sum_{m=1}^{M} \left( \frac{\partial \mathcal{L}^{UANA}}{\partial \hat{f}_m} \times \frac{\partial \hat{f}_m}{\partial \mathbf{x}} \right) \tag{4}$$

where $\hat{f}_m$ represents one of the networks in the ensemble. Figure 1(a) depicts this collective procedure where each individual network in the ensemble votes for the direction where updating the design will lead to the maximal accuracy and minimal uncertainty.

## 3.2 Uncertainty aware tandem (`UA-tandem`)

`Tandem` is a representative of architecture-based methods in which we train an inverse network $\hat{f}^{-1}(\cdot)$ in a manner resembling the encoder-decoder architecture ([32–34]). Unlike the encoder-decoder approach, we start with training the forward model $\hat{f}(\cdot)$. Then we freeze the trainable parameters of $\hat{f}(\cdot)$ and train the inverse model $\hat{f}^{-1}(\cdot)$ in the position of the encoder in order to decrease the cost function:

$$\mathcal{L}^T(\hat{f}^{-1}(\mathbf{y}^*)) := \arg \min_{\hat{f}^{-1}(\cdot)} \left\| \hat{f}(\hat{f}^{-1}(\mathbf{y}^*)) - \mathbf{y}^* \right\|_2^2. \tag{5}$$

Once $\hat{f}^{-1}(\cdot)$ is trained we can simply query it to find designs with our desired performances:

$$\hat{f}^{-1}(\mathbf{y}^*) = \mathbf{x}. \tag{6}$$

The uncertainty-aware tandem (`UA-tandem`) follows the same procedure except that it replaces $\hat{f}(\cdot)$ with $\mathbb{F}_\mu(\cdot)$. Additionally, it includes the uncertainties in the loss:

$$\mathcal{L}^{UAT}(\hat{f}^{-1}(\mathbf{y}^*)) := \arg \min_{\hat{f}^{-1}(\cdot)} \left\| \mathbb{F}_\mu(\hat{f}^{-1}(\mathbf{y}^*)) - \mathbf{y}^* \right\|_2^2 + \alpha \, \mathbb{F}_{\sigma A}(\hat{f}^{-1}(\mathbf{y}^*)) + \beta \, \mathbb{F}_{\sigma E}(\hat{f}^{-1}(\mathbf{y}^*)). \tag{7}$$

Figure 1(b) depicts the architecture of `UA-tandem`.

## 3.3 Predictive uncertainty using Deep Ensembles

Deep Ensembles comprises of an ensemble of $M$ neural networks $\hat{f}_m$ each capable of a prediction $\mu_m(\mathbf{x})$ and its associated uncertainty $\sigma_m(\mathbf{x})$ in form of a Gaussian distribution $\mathcal{N}(\mu_m(\mathbf{x}), \sigma_m(\mathbf{x}))$. The cost function for training each network in the ensemble is the negative log likelihood [29]:

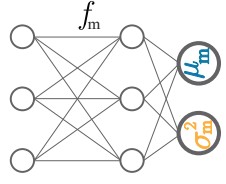

$$\mathcal{L}_m^{NLL} := \frac{\log(\sigma_m^2(\mathbf{x}))}{2} + \frac{(\mathbf{y}^* - \mu_m(\mathbf{x}))^2}{2\sigma_m^2(\mathbf{x})}. \tag{8}$$

Intuitively, in case of the aleatoric uncertainty $\mu_m(\mathbf{x})$ fails to reliably predict $\mathbf{y}^*$. Hence, $\sigma_m^2(\cdot)$ must increase to reduce the loss while the first term ensures $\sigma_m^2(\cdot)$ does not diverge to infinity.

The next step is to ensemble the results of all the networks into a single prediction and a single uncertainty. Deep Ensembles [22] models the ensemble as a single Gaussian distribution $\mathcal{N}(\mathbb{F}_\mu(\mathbf{x}), \mathbb{F}_\sigma(\mathbf{x}))$ approximating the mixture of $M$ previously computed Gaussian distributions

$$\mathbb{F}_\mu(\mathbf{x}) := \frac{1}{M} \sum_m \mu_m(\mathbf{x}), \tag{9a}$$

$$\mathbb{F}_\sigma(\mathbf{x}) = \frac{1}{M} \sum_m (\sigma_m^2(\mathbf{x}) + \mu_m^2(\mathbf{x})) - \mathbb{F}_\mu^2(\mathbf{x}). \tag{9b}$$

The uncertainty of the ensemble can be decomposed into two input-dependent uncertainties, i.e., aleatoric $\mathbb{F}_{\sigma A}(\mathbf{x})$ and epistemic $\mathbb{F}_{\sigma E}(\mathbf{x})$ through slight modification to Equation 9b [18].

$$\mathbb{F}_\sigma(\mathbf{x}) := \alpha\, \mathbb{F}_{\sigma A}(\mathbf{x}) + \beta\, \mathbb{F}_{\sigma E}(\mathbf{x}), \tag{10a}$$

$$\mathbb{F}_{\sigma A}(\mathbf{x}) := \frac{1}{M}\sum_m \sigma_m^2(\mathbf{x}), \tag{10b}$$

$$\mathbb{F}_{\sigma E}(\mathbf{x}) := \frac{1}{M}\sum_m (\mu_m^2(\mathbf{x}) - \mathbb{F}_\mu^2(\mathbf{x})). \tag{10c}$$

This enables us to control the behavior of the neural inversion by tuning the relative importance of these two uncertainties.

## 4  Evaluation

We evaluate the performance of `Autoinverse` through experimenting with the existing neural inverse methods and their uncertainty-aware counterparts. In the paper, we focus on `NA` and `UANA` while `tandem` and `UA-tandem` are evaluated mainly in the Appendix. We perform the evaluation on a set of applications in robotics, and fabrication-oriented computational design. We report the error by running the experiments 3 times to capture the variations. More details are provided in each case study and in Appendix (Section A).

Equations 3 and 7 have two hyperparameters ($\alpha$ and $\beta$) that keep the balance between the MSE, the aleatoric uncertainty and the epistemic uncertainty. We observe that with a relatively larger epistemic weight ($\beta$) we obtain better results. Exploiting this intuition, we tune these parameters for 3 different sets of values for $\{\alpha, \beta\}$: $\{\{0.1, 1\}, \{1, 10\}, \{10, 100\}\}$. We then make two finer step depending on the outcome of the former evaluation and choose the best set of weights. We keep this budget of 5 experiment runs for the rival methods as well. Typically, we use 10% of the target performance for tuning our inverse methods.

### 4.1  Experiments

**Multi-joint robot**   is a simple inverse kinematics problem which is being used as a standard test for neural inverse problems [21, 30]. In this problem, the *design* is the 1D position of the base of the multi-joint robot along with the 1D rotation of its three joints. The inverse problem concerns finding a combination of positions and angles for the base and the joints such that the tip of the arm lands the target position, i.e., the *performance*. We follow [30] for setting up this experiment and its corresponding analytical equation used as the NFP. The training data consists of 10,000 pairs of samples generated by randomly sampling the NFP.

**Spectral printer**   Spectral printing enables digital fabrication of the object's appearance faithfully ([14, 5]). Unlike reproduction of the color (e.g., RGB), reproducing the spectrum ensures that the original and the duplicate remain visually similar independent of the color of the light source. Spectral printing has various important applications specially in fine art reproduction using both 2D and 3D printing [27, 1, 32]. Deep neural networks are becoming the main computational tool for modeling the spectral printing process specially when dealing with a large number of inks. The final objective is to find the correct ink densities at each pixel that can best estimate the target 31D spectrum.

In this experiment, we create the NFP by simulating a printer using an ensemble of 20 neural networks. The design space comprises the ink densities and our `spectral printer` NFP predicts the spectrum of the resulting color. We use real, measured data from an 8-channel printer with 8 EPSON inks including Cyan (C), Magenta (M), Yellow (Y), Black (K), Light black (LK), Light light black (LLK), Light Cyan (LC), and Light Magenta (LM). The light inks, added to the standard CMYK to improve the print quality, introduce significant multi-modality. All networks in the ensemble NFP are trained on 40,000 printed patches [1] consisting of different ink-density combinations and their corresponding spectra. The ensembling is intended for an accurate NFP and is independent of our use of Deep Ensembles for computing the uncertainty. The visual nature of `spectral printer` makes it attractive for analyzing different methods. We release this NFP to the public to add another example to the neural inversion testbed.

Table 1: The NFP and surrogate errors (mean $\pm$ STD) of different neural inverse methods on `multi-joint robot` for 1000 target locations.

| Error | NA | UANA | tandem | UA-tandem | MINI | INN |
|---|---|---|---|---|---|---|
| NFP | $(3.24 \pm 0.51)$ $\times 10^{-4}$ | $\mathbf{(3.21 \pm 1.48)}$ $\mathbf{\times 10^{-6}}$ | $(4.42 \pm 1.56)$ $\times 10^{-3}$ | $\mathbf{(4.04 \pm 0.38)}$ $\mathbf{\times 10^{-5}}$ | $1.6$ $\times 10^{-3}$ | $(9.48 \pm 0.021)$ $\times 10^{-3}$ |
| Surrogate | $(1.99 \pm 0.05)$ $\times 10^{-8}$ | $(9.13 \pm 6.08)$ $\times 10^{-7}$ | $(8.58 \pm 3.00)$ $\times 10^{-6}$ | $(7.10 \pm 0.64)$ $\times 10^{-6}$ | $0$ | $(2.04 \pm 0.017)$ $\times 10^{-13}$ |

**Soft robot** are made of soft, flexible materials. This unique property has made them suitable candidates for interaction with humans in applications like minimally invasive surgery and advanced prosthetics [7]. Unlike `multi-joint robot` with a limited number of rotating joints, each segment of the soft robot is a potential actuator that through their contraction and expansion can determine the robot's final shape. This inverse kinematics problem is typically solved through partial differential equations ([38]). In order to accelerate the solve time of this inverse kinematics problem, Sun et al. [33] proposed a neural surrogate modeling of the problem and its inversion via `tandem`. The design space in this problem consists of the contraction or expansion of 40 controllable soft edge segments. The superposition of all the actuations determines the final deformation position of the soft robot via the position of its 206 vertices.

We use an FEM-based simulation ([38, 15]) as our NFP. We model the relationship between the actuations and the final shape of the soft robot with a neural-network surrogate. Our goal is to solve the neural inversion to find a suitable set of actuations (design) that brings the tip of the soft robot to the target position (performance). The training data consists of 50,000 samples queried by random sampling the actuation with an expansion ratios between -0.2 and 0.2 [33]. The designs are then evaluated on the FEM-based NFP to calculate their deformations.

### 4.2 Quantitative comparison of different neural inversion methods

We evaluate the accuracy of a set of neural inversion methods on `multi-joint robot` in terms of both the NFP and the surrogate errors. The surrogate error is the difference between the 're-prediction' of the obtained inverse solution by the surrogate neural network and the target performance. The NFP error is the difference between the target and the performance of the generated design evaluated on the NFP. In addition to the core methods described so far, we evaluate mixed-integer neural inversion (`MINI`) [2], and the invertible neural network (`INN`) [3].

Table 1 summarizes the inversion results on 1000 randomly sampled target locations for the multi-joint robotic arm. We keep our evaluation fair by setting the capacity of the neural surrogates comparable wherever possible. For instance, all methods except `MINI` have around 3 million parameters (see Appendix, Table 3 for more details). We also used equal computational resources for the tuning. Methods with hyperparameters, like `UANA` and `UA-tandem`, are tuned in 5 stages by hand. Alternatively, the methods without hyperparameters (`NA, tandem`) are given $5\times$ extra budget for inversion: We run `NA` and `tandem` $5\times$ and choose the model that generates the best NFP error. We repeat this process $3\times$ and report the standard error. `MILP` and `INN` are fundamentally different methods. `MILP` finds the global optimum and thus does not need tuning. `INN` has a latent space which we can sample to generate diverse designs. We sample `INN`'s latent space 1024 times for all 1000 targets, evaluate them on the NFP, and report the best NFP error. We repeat this process three times to generate the standard error.

Table 1 shows the outstanding accuracy of the inverse methods that adopt `Autoinverse`, i.e., `UANA` and `UA-tandem`, in terms of the NFP error. It also demonstrate how even a perfect surrogate error (e.g., `MINI`) does not guarantee accurate solutions when tested on the NFP. A second look at Table 1 reveals further interesting insights. Although `NA` obtains notably lower surrogate error than its uncertainty-aware counterpart (`UANA`), it performs significantly worse in terms of the NFP error. The main reason for this trend is that, when optimizing the surrogate, `UANA` is not only concerned with finding accurate designs leading to a small accuracy gap between the target and candidate performances, i.e., the surrogate error, but also with those designs featuring low uncertainty (through the uncertainty term in Equation 10). Therefore, `UANA` achieves high accuracy in terms of the essential NFP error at the cost of worsening the inconsequential surrogate error. Furthermore, `UA-tandem`, for example, achieves better performance than `NA`. This is a remarkable result for an architecture-based

Table 2: The distribution of ink densities ($\geq 0.4$) after the inversion of `spectral printer` using `UANA`. Once we insert noise into LC channel or sample it sparsely, `Autoinverse` detects and avoids it. STD has been rounded to the nearest integer.

| Model | dataset | NFP error | C | M | Y | K | LC | LM | LK | LLK |
|-------|---------|-----------|---|---|---|---|----|----|----|-----|
| `UANA` | Standard | $(6.30 \pm 0.031) \times 10^{-3}$ | $186 \pm 2$ | $67 \pm 4$ | $63 \pm 6$ | $3 \pm 0$ | $437 \pm 2$ | $356 \pm 7$ | $26 \pm 3$ | $348 \pm 5$ |
| | Sparse | $(5.64 \pm 0.017) \times 10^{-3}$ | $316 \pm 1$ | $60 \pm 2$ | $59 \pm 1$ | $2 \pm 0$ | $\mathbf{0 \pm 0}$ | $326 \pm 5$ | $25 \pm 4$ | $323 \pm 11$ |
| | Noisy | $(6.13 \pm 0.026) \times 10^{-3}$ | $263 \pm 1$ | $67 \pm 4$ | $34 \pm 2$ | $1 \pm 0$ | $\mathbf{0 \pm 0}$ | $276 \pm 3$ | $29 \pm 3$ | $378 \pm 5$ |
| `NA` | Standard | $(1.57 \pm 0.001) \times 10^{-1}$ | $1895 \pm 13$ | $1060 \pm 26$ | $1795 \pm 18$ | $162 \pm 11$ | $865 \pm 8$ | $1396 \pm 30$ | $179 \pm 15$ | $2378 \pm 19$ |
| | Sparse | $(1.34 \pm 0.007) \times 10^{-1}$ | $905 \pm 6$ | $606 \pm 7$ | $1515 \pm 8$ | $137 \pm 6$ | $1604 \pm 11$ | $2130 \pm 17$ | $294 \pm 15$ | $2312 \pm 24$ |
| | Noisy | $(1.47 \pm 0.002) \times 10^{-1}$ | $1192 \pm 14$ | $988 \pm 10$ | $1128 \pm 10$ | $55 \pm 6$ | $1029 \pm 23$ | $948 \pm 21$ | $283 \pm 12$ | $1742 \pm 37$ |

Figure 2: Distribution of actuation values for `soft robot`. Edge 16 is corrupted by noise (top) and sampled sparsely (bottom) in positive range. While other randomly chosen edges feature both negative and positive actuations, `UANA` produces solutions without (or with less) positive actuations for Edge 16.

method given it is significantly faster than the optimization-based `NA`. We report extensively the time performance and the details of the training for this and the next experiments in the Appendix.

## 4.3 Neural inversion in the presence of imperfect data

One of the main advantages of `Autoinverse` appears in scenarios where the training data suffers from noise, e.g., measurement noise or poor sampling, e.g., sparsity in some regions. We evaluate the performance of `Autoinverse` on imperfect training data using `NA` and `UANA` (`tandem` and `UA-tandem` are evaluated under the same configuration in the Appendix Section C).

**Locally sparse data**   We sample both `soft robot` and `spectral printer` NFPs (Section 4.1) in a way that the data does not contain any samples from one of the inputs in a specific interval. For `spectral printer`, we would like to find ink densities to reproduce the spectra of the colors in the painting in Figure 3 (made of 3568 distinct color spectra). We sample the printer channels at 0 (no ink), 0.05, 0.1, 0.5, and 1 (full ink) densities to form the *standard* training data (with no known uncertainty). We create a partially *sparse* dataset similar to the standard one except for the Light Cyan (LC) channel for which we only have samples at 0, 0.05 and 0.1. Table 2 shows that while for the standard dataset `UANA` finds inverse solutions that include the LC channel frequently (437 times), for the sparse dataset it avoids this channel *completely* and compensate for it using the Cyan channel. `UANA` is able to avoid this channel as the epistemic uncertainty increases in sparse regions of dataset (see Appendix Section C.4 for more details).

In `soft robot` we sample the 16th (among 40) controllable edge only in the negative range (contraction only). We then use the trained network to invert 1000 test samples. Figure 2 shows the distribution of each edge for 1000 inversion tasks. We have plotted the distribution for the 16th edge as well as for 7 other randomly chosen edges. As evident from Figure 2 bottom row, `UANA` is highly reluctant to choose designs with positive actuations for this edge.

**Locally noisy dataset**   With the same problem configuration as before, we would like to test the robustness of `Autoinverse` on a dataset locally corrupted with noise. We start with a standard dataset and inject Gaussian noise $\mathcal{N}(0, 0.1)$ to the spectrum of the samples with more than $0.4$ LC density. Table 2 shows how after introducing noise to the LC, the network avoids that channel and compensates it by using more Cyan instead. In `soft robot` we corrupt the final shape of the soft

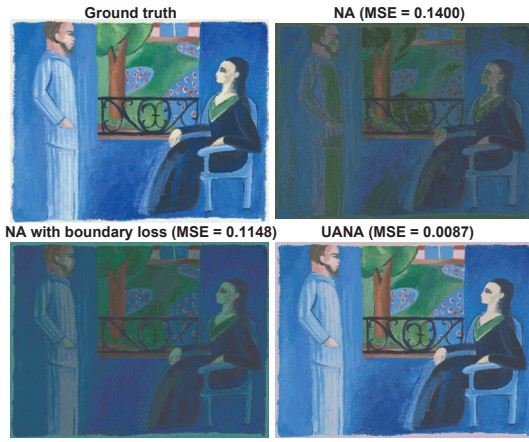

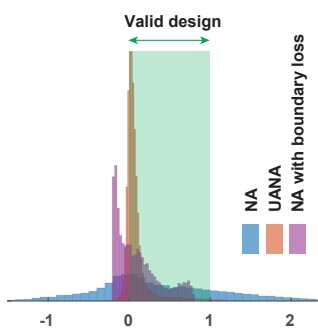

Figure 3: In painting reproduction, `UANA` outperforms `NA` with and without a boundary loss term. The distribution of the inverse solutions is shown on right.

robot with Gaussian noise $\mathcal{N}(0, 0.5)$ for all the shapes where the 16th edge has positive actuation. As we can see in Figure 2 (top row), the inversion has completely avoided positive actuations for this edge.

### 4.4 Autoinverse brings AutoML to neural inversion

**Autoinverse incorporates feasibility.** Deep Ensembles produces high epistemic uncertainty outside the distribution of the training data. This includes the regions where the NFP is not defined and thus not sampled. For example, ink densities outside $[0, 1]$ are not printable. If such cases arise during the inversion, they are clipped to $[0, 1]$ [1]. In such regions, networks in the ensemble do not agree and thus the epistemic uncertainty increases. `Autoinverse` automatically avoids these unfeasible regions. In order to simulate the result of the inversion, after clipping the out-of-range densities we feed them into the NFP. As evident from Figure 3 left, inversion via NA results in a poor reproduction of the original painting. `UANA`, on the other hand, achieves spectacular reproduction accuracy. This is explained by the plot in Figure 3 right, showing the distribution of ink densities obtained from both methods.

**Autoinverse has built-in regularization.** We incorporate regularization into inversion methods in order to obtain solutions that, among other purposes, agree with the observations and follow a certain statistical distribution. Oftentimes, regularization is case-specific, requires human knowledge, and comes with unexpected side effects. Here we show that `Autoinverse` follows the distribution of the training data naturally by taking into account the epistemic uncertainty. We validate this point using both `spectral printer` and `soft robot` experiments. We show that `Autoinverse` without any explicit regularization performs better or on par with its counterpart inversion methods equipped with regularization.

In the `spectral printer` experiment, we evaluate the effect of the *boundary loss*, originally proposed as a generic regularization for `NA` to limit the designs within a box constraint (see [30] and Appendix Section D). The boundary loss is added to Equation 1 and weighted using a hyperparameter. We tune this parameter with the same tuning budget we allocate for tuning the uncertainty weights (5 set of inversions on evaluation data). In Figure 3, we observe that although `NA` with boundary loss improves the distribution of ink densities within the valid region ($[0, 1]$), it still trails the regularization-free `UANA` significantly.

Regularizing the `soft robot` is less intuitive as the superposition of all actuations determines the final shape and whether it is physically plausible. In [33], the regularization is a smoothness term that keeps actuation values near each other (see Appendix Section D). Figure 4(a)(1) demonstrates how `NA` fails without regularization to control the robot with a reasonable deformation. Once the regularization is added to `NA`, the designs become physically meaningful (Figure 4(a)(2)). Figure 4(a)(3) shows how `UANA` performs comparably without any regularization.

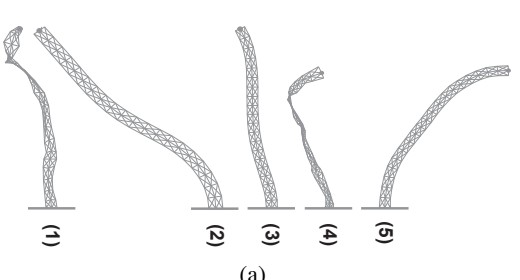
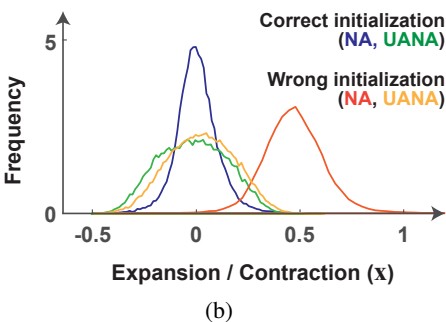

Figure 4: On the left we can see the distribution of actuations calculated by `UANA` and `NA` with two different initialization: one initialization near the training data distribution and one far from it. On the right we can see a range of randomly chosen soft robot shapes calculated by different methods with different regularization and initialization.

**Autoinverse is initialization-free.** Sensitivity to the initialization is a widely known issue in non-convex optimizations. Despite equipping `NA` with the smoothness term in `soft robot`, Figure 4(b) demonstrates how an incorrect initialization can result in solutions with seemingly good surrogate and regularization loss (see Appendix Section D) but in the infeasible region of the design space (Figure 4(a)(4)). `UANA` with the same incorrect initialization *and* without any regularization, leads to soft robot designs that reach the target location accurately and produce plausible deformations (Figure 4(a)(5)). For `UANA`, the solver starts with reducing the main contributor to the optimization objective, i.e., the epistemic uncertainty. Once a region with a reasonably small uncertainty is reached, the accuracy term starts to take effect and a desirable solution within the valid range of the design space is found.

### 4.5 Ablation studies

**Separating the ensembling effect**  Although in both `NA` and `UANA` we use surrogates with similar capacity (network size), one could argue that the higher performance of `UANA` comes from the ensemble architecture of its surrogate. We detach the impact of ensembling and uncertainty awareness on the inversion performance by implementing `NA ensemble` where, instead of a single, large forward neural network, it uses an ensemble of networks. The inversion procedure is identical to `NA` (see Appendix Section E and Figure 5). We employ `NA ensemble` in `multi-joint robot` with the same configurations as in Section 4.2. The surrogate and NFP error for `NA ensemble` are $(3.30 \pm 0.59) \times 10^{-9}$ and $(1.17 \pm 0.32) \times 10^{-4}$, respectively. Comparing these values with those in Table 1, `NA ensemble` shows a slight improvement over `NA` but is significantly outperformed by `UANA`.

**Diversity of activation functions in Deep Ensembles**  Deep Ensembles uses a similar set of networks with identical activation functions [22]. In practice, we observe that a diverse set of activation layers leads to a better performance of `Autoinverse`. Different activation functions generate different behaviours, show higher disagreement where the training data is under-represented and, thus, result in an accurate estimation of the landscape of epistemic uncertainty [36]. For all experiments in this work, we use a diverse range of activation functions (see Appendix Section A). As an ablation, we run `UANA` on `spectral printer` using sparse data (same configuration as Section 4.3) but with ReLU as the only activation layer. In contrast to Table 2, `UANA` (with ReLU only activation) does not completely avoid the sparse domain and delivers $19 \pm 4$ solutions that contain LC with densities $\geq 0.4$.

## 5 Discussion

The `Autoinverse` cost function is multi-objective. Instead of finding a single solution through a weighted combination of these objectives (what we have seen so far), we can capture the trade-off between the accuracy and uncertainty through computing the Pareto front (see Appendix Section

F for more details). `Autoinverse` is an inversion *strategy* that could be applied to various inverse methods, especially those that concern imitation of an original process (dubbed as NFP in our work). We look forward to see `Autoinverse` adopted for more inverse architectures beyond `tandem`, and more direct optimizations beyond first-order `NA`.

Predicting the uncertainty using Deep Ensembles [22] is not the most efficient solution. This is because multiple forward networks are trained in two stages (once for the mean, and once jointly for the mean and variance). It is highly interesting to integrate more efficient uncertainty estimation methods, such as Monte Carlo dropout [11, 18], into `Autoinverse`.

## Acknowledgments and Disclosure of Funding

We are grateful to Xingyuan Sun and Szymon Rusinkiewicz who provided the repository of the soft robot simulation [33], Sebastian Cucerca for the revision of the figures, and anonymous reviewers for valuable feedback. Special thanks to Jalil Maani and Soraya Iranpak for their kind support.

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
