# Appendix

## A   Standard error calculation, tuning, and implementation details

In this section we provide more details on different inverse methods we use in the paper. As many of the inverse methods we use in the paper have a stochastic component, we perform all following experiments 3 times and report the standard error. Since we have a budget of 5 runs for tuning the hyperparameters of `Autoinverse` methods, we allocate similar or higher resources for other methods in order to make the comparisons fair.

`NA`   To avoid local minima, we run 50 *solves* of inversion with random initialization (each having up to 2000 iterations). As an alternative to tuning, we run `NA` 5 times and report the best NFP error. To accelerate the inversion in this paper we perform batch optimization. Thus, we can increase the number of target samples (up to GPU memory limit) without impacting the inversion time noticeably.

`NA ensemble`   The configurations for `NA ensemble` is similar to `NA` except for the forward model. Unlike `NA`, we have an ensemble of networks ($\mathbb{F}_\mu$) that comprises the forward model of `NA ensemble`:

$$\mathbb{F}_\mu(\mathbf{x}) := \frac{1}{M} \sum_m \mu_m(\mathbf{x}). \tag{11}$$

Unlike `UANA`, single networks in the ensemble are incapable of predicting uncertainty. The cost function for `NA ensemble` is therefore defined as:

$$\mathcal{L}^{NA_{en}}(\mathbf{x}) := \arg\min_x \|\mathbb{F}_\mu(\mathbf{x}) - \mathbf{y}^*\|_2^2 \tag{12}$$

Similar to `UANA`, the back propagation is based on an ensemble of gradients coming from all the single networks in the ensemble (Figure 5).

$$\mathbf{x}^z = \mathbf{x}^{z-1} - \delta \sum_{m=1}^{M} \left( \frac{\partial \mathcal{L}^{NA_{en}}}{\partial \hat{f}_m} \times \frac{\partial \hat{f}_m}{\partial \mathbf{x}} \right) \tag{13}$$

Similar to `NA`, we run `NA ensemble` 5 times and pick the best results with respect to the NFP error.

`tandem`   Since `tandem` does not explicitly possess any hyperparameter, instead of tuning, we evaluate 5 pre-trained inverse models (each initialized differently) and select the one that has the best NFP error on 10% of the target data. We then perform the inversion on the remaining target data using the best-performing inverse model.

`UANA` **and** `UA-tandem`   In Section 4.5 of paper we showed how using a diverse range of activation functions in the ensemble network improves the quality of epistemic uncertainty and consequently the results of the inversion. For training the surrogates used in `UANA`, `UA tandem` and `NA ensemble` we use 10 networks in the ensemble with the following activations $\mathbb{F}_\mu$:

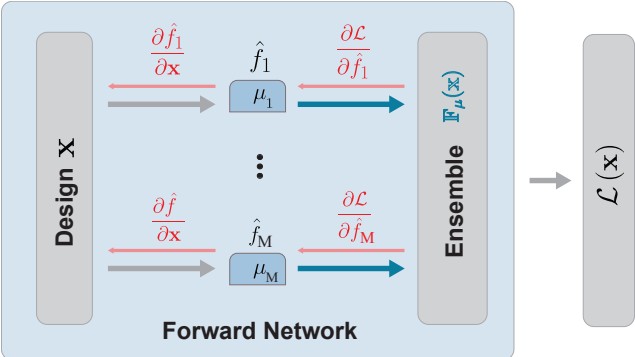

Figure 5: NA ensemble architecture

Table 3: Training details of different neural surrogate models used in inverse methods for `multi-joint robot`.

| Network's name | Sub-networks name | Trainable parameters | Layer configuration |
|---|---|---|---|
| INN | - | 3727416 | [30, 3] |
| NA | - | 3204302 | 100, 1000, 1500, 1000, 100 |
| NA ensemble | Forward networks | $351802 \times 10$ | 100, 500, 500, 100 |
| UANA | $\mu$ networks | $351802 \times 10$ | 100, 500, 500, 100 |
| | $\sigma$ networks | $20902 \times 10$ | 100, 100, 100 |
| Tandem | Forward network | 3204302 | 100, 1000, 1500, 1000, 100 |
| | Inverse network | 113804 | 100, 250, 250, 100 |
| UA-tandem | $\mu$ networks | $351802 \times 10$ | 100, 500, 500, 100 |
| | $\sigma$ networks | $20902 \times 10$ | 100, 100, 100 |
| | Inverse network | 117108 | 100, 250, 250, 100 |
| MINI | - | 10802 | 100, 100 |

- Tanh $\times 2$
- ReLU $\times 2$
- CELU $\times 2$
- LeakyReLU $\times 2$
- ELU
- Hardswish

We use ReLU activation functions for all other methods.

INN  For each one of the 1000 target performances we randomly sample the latent space of the INN architecture [30] 1024 times. Thus, we end up with 1024 designs (for a single target) and evaluate all designs on the NFP and report the best error as the NFP error. For the surrogate error, we report the average forward loss.

MINI  MINI is based on a mixed-integer optimization which is capable of finding the globally optimum solution [2]. This method is deterministic and every run returns the same solution, as a result we do not report the standard error for MINI.

**Hardware**  To have a fair comparison we run all the methods on the same GPU machine for evaluating time performances. We used an NVIDIA TITAN X GPU for time evaluation. For other evaluations we used a GPU cluster. Training the forward models is trivially parallelizable. Moreover, we can parallelize 50 iterations of NA and UANA and aggregate the data in a post-processing step and choose the best results *based on the surrogate error*. Nevertheless, we are reporting our computation time assuming both training and inversion are performed serially on a single GPU.

## B    Training details for neural surrogate models in `multi-joint robot` (Table 1 in paper)

In Table 3 and Table 4, we can compare the capacity, training time, and accuracy of the neural network surrogates used for the inversion of `multi-joint robot`. We keep similar training capacity for all methods except MINI. MINI uses a combinatorial optimization and is not scalable to large networks [2]. We trained a smaller surrogate for MINI but at the same time we monitored its training loss to lie within a reasonable range.

## C    Details for 'Neural inversion in the presence of imperfect data'

### C.1    **Counterpart results of** `tandem` **and** `UA-tandem` **for** `spectral printer`

Table 5 presents the results of `spectral printer`, similar to Section 4.3 (Table 2) of the paper but comparing `tandem` and `UA-tandem`. As evident from the table, we obtain similar performance gain

Table 4: [Continued] Training details of different neural surrogate models used in inverse methods for `multi-joint robot`.

| Network's name | Sub-networks name | Total training time (s) | Total inversion time (s) | Training loss |
|---|---|---|---|---|
| INN | - | - | $1024 \times (1.3 \times 10^{-2})$ | $2.10 \times 10^{-2}$ |
| NA | - | 190 | 374 | $3.33 \times 10^{-5}$ |
| NA ensemble | single | 1650 | 426 | $3.91 \times 10^{-6}$ |
| UANA | $\mu$ networks | 1650 | 1075 | $2.34 \times 10^{-6}$ |
| | $\sigma$ networks | 192 | | |
| Tandem | Forward network | 190 | $3.8 \times 10^{-3}$ | $3.33 \times 10^{-5}$ |
| | Inverse network | 181 | | |
| UA-tandem | $\mu$ networks | 1650 | $3.8 \times 10^{-3}$ | $2.34 \times 10^{-6}$ |
| | $\sigma$ networks | 192 | | |
| | Inverse network | 1109 | | |
| MINI | - | 150 | $2.98 \times 10^4$ | $4.89 \times 10^{-4}$ |

Table 5: The distribution of ink densities ($\geq 0.4$) after the inversion of `spectral printer` using `UA-tandem`. Once we insert noise into LC channel or sample it sparsely, `Autoinverse` detects and avoids it. STD has been rounded to nearest integer.

| Model | Data set | NFP error | C | M | Y | K | LC | LM | LK | LLK |
|---|---|---|---|---|---|---|---|---|---|---|
| UA-tandem | Standard | $(2.62 \pm 0.488) \times 10^{-3}$ | $180 \pm 12$ | $48 \pm 2$ | $13 \pm 4$ | $3 \pm 1$ | $174 \pm 8$ | $36 \pm 12$ | $0 \pm 0$ | $0 \pm 0$ |
| | Sparse | $(2.34 \pm 0.097) \times 10^{-3}$ | $291 \pm 11$ | $43 \pm 0$ | $15 \pm 2$ | $3 \pm 1$ | $\mathbf{0 \pm 0}$ | $89 \pm 16$ | $0 \pm 0$ | $0 \pm 0$ |
| | Noisy | $(5.16 \pm 0.423) \times 10^{-3}$ | $242 \pm 2$ | $63 \pm 0$ | $18 \pm 1$ | $1 \pm 0$ | $\mathbf{0 \pm 0}$ | $64 \pm 32$ | $8 \pm 1$ | $29 \pm 2$ |
| tandem | Standard | $(3.51 \pm 2.903) \times 10^{-2}$ | $159 \pm 9$ | $40 \pm 10$ | $0 \pm 0$ | $12 \pm 9$ | $255 \pm 124$ | $187 \pm 256$ | $5 \pm 4$ | $1070 \pm 1514$ |
| | Sparse | $(2.18 \pm 0.840) \times 10^{-2}$ | $208 \pm 22$ | $36 \pm 3$ | $0 \pm 0$ | $17 \pm 7$ | $52 \pm 30$ | $20 \pm 9$ | $14 \pm 19$ | $0 \pm 0$ |
| | Noisy | $(3.82 \pm 0.810) \times 10^{-2}$ | $174 \pm 29$ | $60 \pm 32$ | $39 \pm 42$ | $71 \pm 9$ | $303 \pm 171$ | $192 \pm 239$ | $114 \pm 123$ | $62 \pm 74$ |

when augmenting `tandem` with uncertainty awareness. Similar to `UANA`, `UA-tandem` has completely avoided the LC channel.

## C.2 Soft robot actuation distribution

Figure 2 in the paper showed the actuation distribution of only 8 soft robot edges (Section 4.3 in paper). In Figure 6 we show the distribution of actuation for *all* edges computed by `UANA` on surrogates learned using partially noisy and partially sparse data. We repeat this experiment using `UA-tandem` to emphasize on the generality of `Autoinverse` (Figure 7).

## C.3 Training details of surrogates used for `spectral printer` and `soft robot`

Tables 6 and 7 show the layer configuration, number of trainable parameters, the training and inversion time, and the training loss of different surrogate models used in `spectral printer`. Tables 8 and 9 show the layer configuration, number of trainable parameters, the training and inversion time, and the training loss of different surrogate models used in `soft robot`. From the tables we can observe that

Table 6: Training details of different neural surrogate models used in inverse methods for `spectral printer`.

| Network's name | Sub-networks name | Trainable parameters | Layer configuration |
|---|---|---|---|
| NA | - | 905931 | 100, 500, 800, 500, 100 |
| NA ensemble | Forward networks | $64531 \times 10$ | 100, 100, 200, 100, 100 |
| UANA | $\mu$ networks | $64531 \times 10$ | 100, 100, 200, 100, 100 |
| | $\sigma$ networks | $24231 \times 10$ | 100, 100, 100 |
| Tandem | Forward network | 905931 | 100, 500, 800, 500, 100 |
| | Inverse network | 117108 | 100, 250, 250, 100 |
| UATandem | $\mu$ networks | $64531 \times 10$ | 100, 100, 200, 100, 100 |
| | $\sigma$ networks | $24231 \times 10$ | 100, 100, 100 |
| | Inverse network | 117108 | 100, 250, 250, 100 |

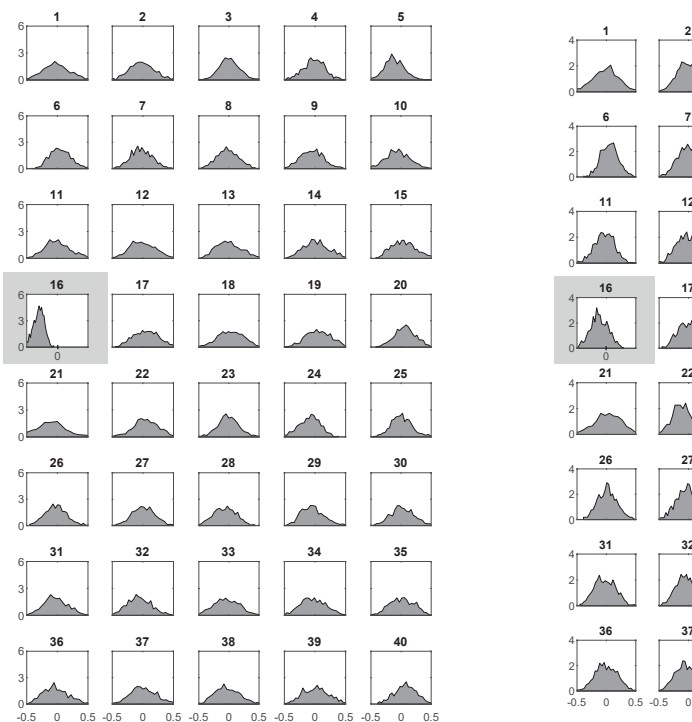

(a) 16th edge is noisy in the positive actuation range.

(b) The training data does not contain any samples with positive actuation of the 16th edge.

Figure 6: The actuation distribution of all the edges for inversion on both noisy and sparse data via `UANA`.

Table 7: [Continued] Training details of different neural surrogate models used in inverse methods for `spectral printer`.

| Network's name | Sub-networks name | Training time | Inversion time | Training loss |
|---|---|---|---|---|
| NA | - | 295 | 300 | $4.45 \times 10^{-6}$ |
| NA ensemble | Forward networks | 441 | 563 | $3.38 \times 10^{-6}$ |
| UANA | $\mu$ networks | 441 | $1.15 \times 10^3$ | $3.44 \times 10^{-6}$ |
|  | $\sigma$ networks | 240 |  |  |
| Tandem | Forward network | 295 | $1.07 \times 10^{-2}$ | $4.45 \times 10^{-6}$ |
|  | Inverse network | 260 |  |  |
| UATandem | $\mu$ networks | 441 |  |  |
|  | $\sigma$ networks | 240 | $1.04 \times 10^{-2}$ | $3.44 \times 10^{-6}$ |
|  | Inverse network | $1.08 \times 10^3$ |  |  |

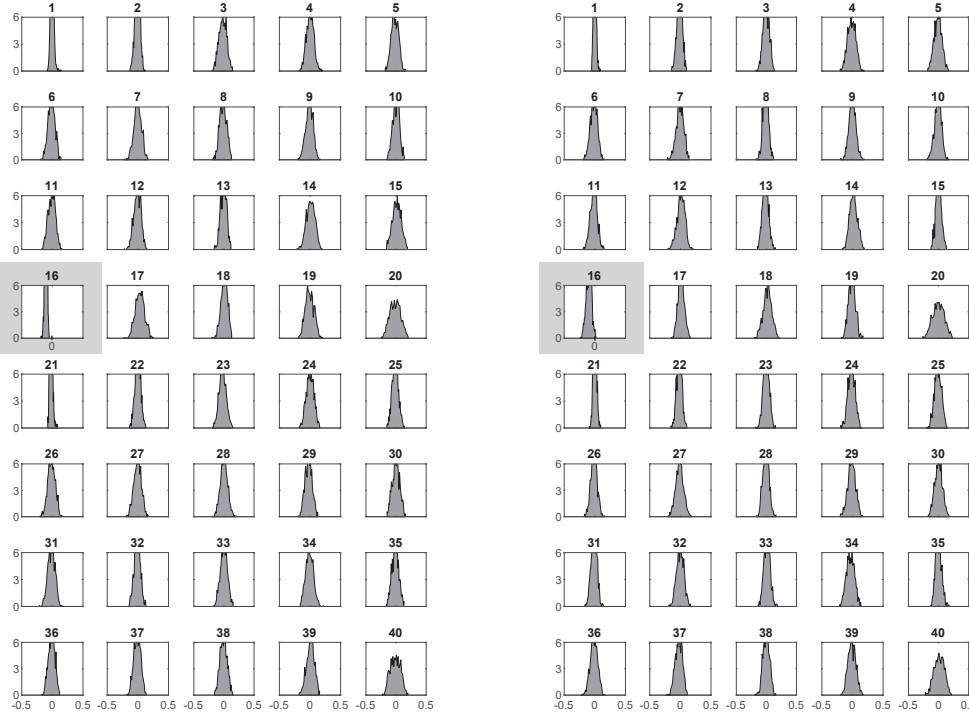

(a) 16th edge is noisy in the positive actuation range.

(b) The training data does not contain any samples with positive actuation of the 16th edge.

Figure 7: The actuation distribution of all the edges for inversion on both noisy and sparse data via `UA-tandem`.

Table 8: Training details of different neural surrogate models used in inverse methods for `soft robot`.

| Network's name | Sub-networks name | Trainable parameters | Layer configuration |
|---|---|---|---|
| NA | - | 45506206 | 2000, 5000, 5000, 2000 |
| UANA | $\mu$ networks | $3227606 \times 10$ | 300, 1500, 1500, 300 |
| | $\sigma$ networks | $45106 \times 10$ | 100, 100, 100 |
| Tandem | Forward network | 45506206 | 2000, 5000, 5000, 2000 |
| | Inverse network | 3227440 | 300, 1500, 1500, 300 |
| UATandem | $\mu$ networks | $3227606 \times 10$ | 300, 1500, 1500, 300 |
| | $\sigma$ networks | $45106 \times 10$ | 100, 100, 100 |
| | Inverse network | 3227440 | 300, 1500, 1500, 300 |

Table 9: [Continued] Training details of different neural surrogate models used in inverse methods for `soft robot`.

| Network's name | Sub-networks name | Total training time (s) | Total inversion time (s) | Training loss |
|---|---|---|---|---|
| NA | - | 5428 | 2250 | $2.63 \times 10^{-4}$ |
| UANA | $\mu$ networks | 16940 | 2950 | $2.39 \times 10^{-5}$ |
| | $\sigma$ networks | 16290 | | |
| Tandem | Forward network | 5428 | $2.90 \times 10^{-1}$ | $2.63 \times 10^{-4}$ |
| | Inverse network | 1740 | | |
| UATandem | $\mu$ networks | 16940 | $1.65 \times 10^{-1}$ | $2.39 \times 10^{-5}$ |
| | $\sigma$ networks | 16290 | | |
| | Inverse network | 9458 | | |

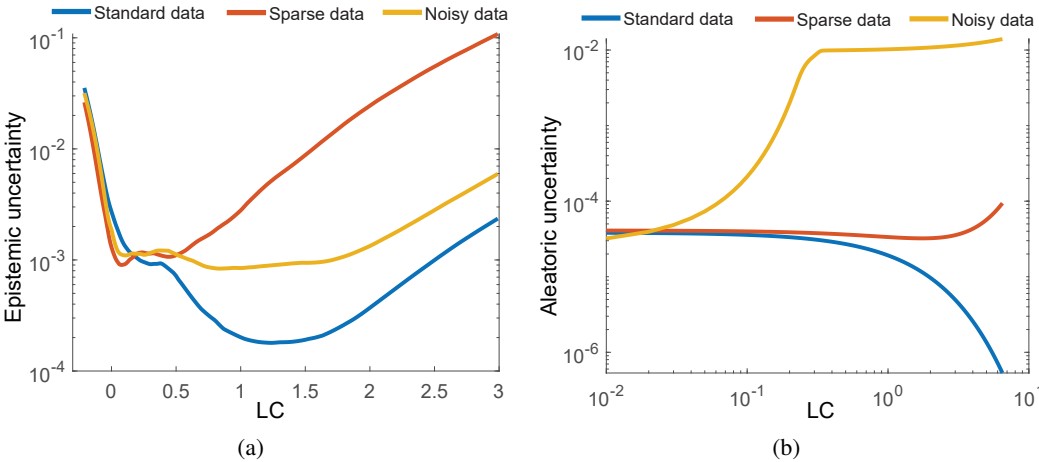

Figure 8: The landscape of the aleatory and epistemic uncertainty.

the training capacity of different surrogates models is comparable. Furthermore, the training accuracy of all models is similar.

### C.4   Epistemic and aleatoric loss behaviour for spectral printing experiment

**Epistemic uncertainty**    The key to handling design feasibility is the epistemic uncertainty (Equation 3(c) paper). We know that the scarcity of data results in higher epistemic uncertainty [18]. On the other hand, by definition, we do not have any infeasible or out-of-range data points in our dataset. Hence, if we query a network with an infeasible or out-of-range input, we will get high uncertainty for the prediction. We use this trend to avoid such samples in the inversion. Figure 8(a) shows the trend of the epistemic uncertainty values for `spectral printer` (Section 4.3, Table 2 in paper). For that experiment, we ran the inversion using `UANA` on 3 different datasets: standard, noisy, and sparse and observed how the problematic ink channel (LC) is avoided. Figure 8(a) demonstrates *why* that ink channel is avoided. In Figure 8(a), we set all ink channels except LC to 0 while increasing the values of LC ink density from 0 (the x-axis of the plot).

As expected, for all three datasets moving away from the feasible region (between 0 and 1) increases the epistemic uncertainty. When trained for the sparse data (red curve), where the LC channel has not been sampled after 0.4, the epistemic uncertainty starts to increase earlier.

Outside the feasible region, each network in the ensemble has to extrapolate as it has not been trained in those regions. Consequently, the predictions of ensemble networks diverge. The divergence of the networks increases the epistemic uncertainty and, during the inversion, the uncertainty aware methods can reject solutions in these regions.

**Aleatoric uncertainty**    Figure 8(b) demonstrates the behavior of the aleatoric uncertainty of the surrogate used for the same experiment (`UANA` on `spectral printer`). Similarly, to generate the plots, we set all the ink channels to 0 and change the values of the Light Cyan ink densities. As evident from Figure 8(b), the level of uncertainty increases significantly for the noisy dataset, while for sparse and standard datasets aleatoric uncertainty is at least two orders of magnitude smaller. The increase of aleatoric uncertainty for the noisy data helps `UANA` avoid any samples from those regions (Tables 2 in the paper and Table 5).

## D   Details for 'Autoinverse brings AutoML to neural inversion'

**Spectral printer**    Complementary to Figure 4 of the paper, in Figure 9 we compare the inversion performance using a diverse range of inverse methods. Here we clearly see that basic methods, such as `NA` and `tandem` fail spectacularly in computing feasible designs.

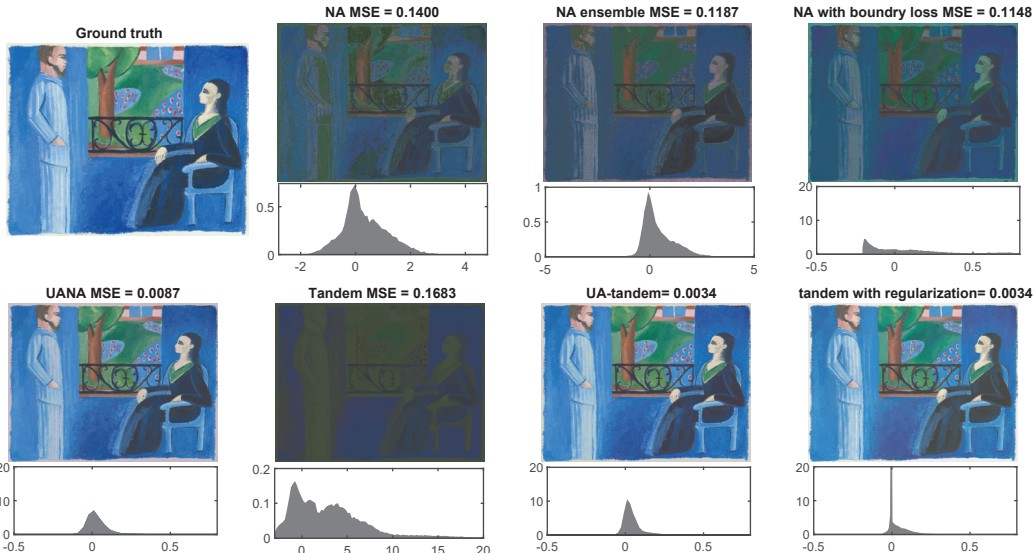

Figure 9: Spectral reproduction of a ground-truth painting using different inverse methods (`spectral printer`). Apart form the reproduction quality, we show the distribution of the computed *designs*, i.e., ink densities. Note that the feasible ink density range is $[0, 1]$.

Boundary loss is a semi-generic regularization, suitable for handling box constraints [30]:

$$\mathcal{L}_{bnd} = \operatorname{Re} LU \left( |x - \mu_x| - \frac{1}{2} R_x \right) \tag{14}$$

where $R_x$ is the value range of design samples in the dataset and $\mu_x$ is their average. Incorporating the boundary loss in `NA` in Figure 9 results in an improvement in the distribution of the ink intensities. However, the results are still far from acceptable.

Note how applying hand-crafted regularization (ink intensity regularization) [32] improves the quality of `tandem` significantly (`tandem with regularization` in Figure 9). `UANA` and `UA-tandem` however perform comparably without any regularization.

**Soft robot objective for the Neural Adjoint method** The objective function for `soft robot` inversion comprises of two terms, one is responsible for bringing the tip of the robot to the target and the other one ($\mathcal{R}(\mathbf{x}^{in})$) guarantees the deformations to remain physical [33].

$$\mathcal{L}(\mathbf{x}^{in}) := \left\| \mathbf{x}_i^{out} - \mathbf{t} \right\|_1 + \lambda \cdot \mathcal{R}(\mathbf{x}^{in})$$
$$i \in [123, \; 124] , \tag{15}$$

$$\mathcal{R}(\mathbf{x}^{in}) := \sum_{\substack{1 < i < n, i \neq n/2, \\ i \neq n/2+1}} \left( \frac{\mathbf{x}_{i+1}^{in} - \mathbf{x}_i^{in}}{2} - \frac{\mathbf{x}_i^{in} - \mathbf{x}_{i-1}^{in}}{2} \right)^2 . \tag{16}$$

where $\mathbf{t}$ represents the target location and $\mathbf{x}_i^{out}$ represents the position of all 206 vertices of soft robot, among which $i \in [123, \; 124]$ represent the position of its tip. Also, $\lambda$ adjusts the importance of the smoothness term and $\mathcal{R}(\mathbf{x}^{in})$ regulates the actuation of the flexible edges ($\mathbf{x}^{in}$ ) to insure that the deformation of the robot is physical.

**Sensitivity to initialization (`soft robot`)** In Section 4.4 of the paper, we learned how hand-crafted regularization improves the quality of the designs. Despite having regularization, `soft robot` inversion using `NA` fails when initialized with values far from feasible region. This is evident from both the irregular robot shapes (Figure 3(b) in the paper) and the distribution of the actuation (Figure 10) computed using **regularized** `NA` but with *wrong* initialization. At the same time, `UANA` without any regularization and with a wrong initialization produces plausible robot shapes (Figure 3(b) in the paper) and actuation distribution (centered around 0). Interestingly, the smoothness score for

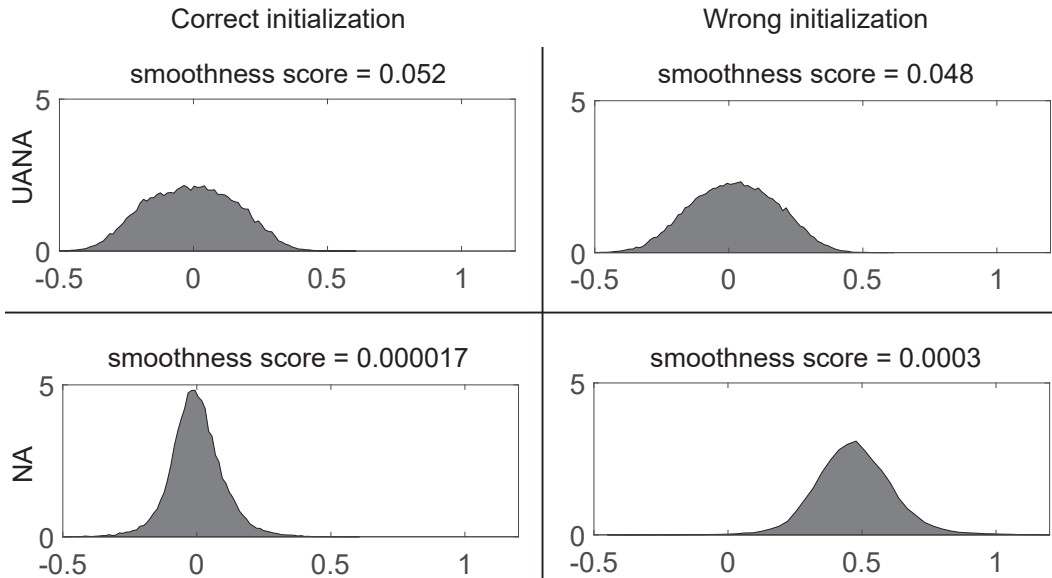

Figure 10: The effect of wrong initialization for inversion of `soft robot` using `NA` with regularization and two different initialization. Note the robustness of `UANA` without any form of regularization.

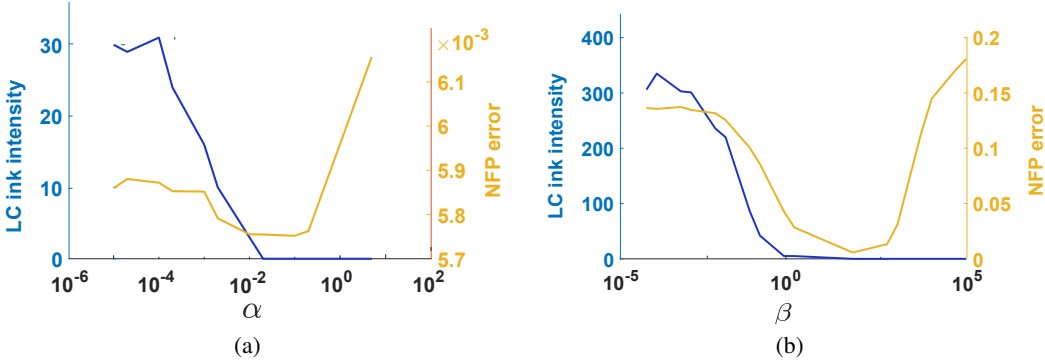

Figure 11: Stability of `Autoinverse` within a wide range of its hyperparameters $\alpha$ and $\beta$.

clearly failed shapes obtained from `NA` with regularization, are for both initializations very small (Figure 10). This shows how even regularization can be multi-modal and fall into a wrong local minima and generate designs with nonphysical shapes (Figure 3(b)).

## E   Details for 'Ablation studies'

As discussed in Section 4.5 of the paper, the power of `UANA` lies in its uncertainty awareness and not the ensembling process. We include the result of `NA ensemble` for paitning reproduction in Figure 9. While we observe a marginal improvement of `NA ensemble` over `NA`, it is clearly outperformed by `UANA`.

**Sensitivity to uncertainty weights**   `Autoinverse` is extremely stable when tuning its hyperparameters, i.e., uncertainty weights ($\alpha$ and $\beta$ in Equation 3 in paper). This ensures that a light hyperparameter tuning is enough for obtaining reasonable results. We evaluate this behavior by using a wide range of $\alpha$ and $\beta$ values spanning over 5 orders of magnitude. This ablation is performed using `UANA`. In the ablation of aleatoric ($\alpha$) and epistemic ($\beta$) weight, we have used the noisy and epistemic data of `spectral printer`, respectively. When evaluating $\alpha$ we keep $\beta$ fixed at the tuned value. Alternatively for the evaluation of $\beta$, $\alpha$ is constant at the tuned value.

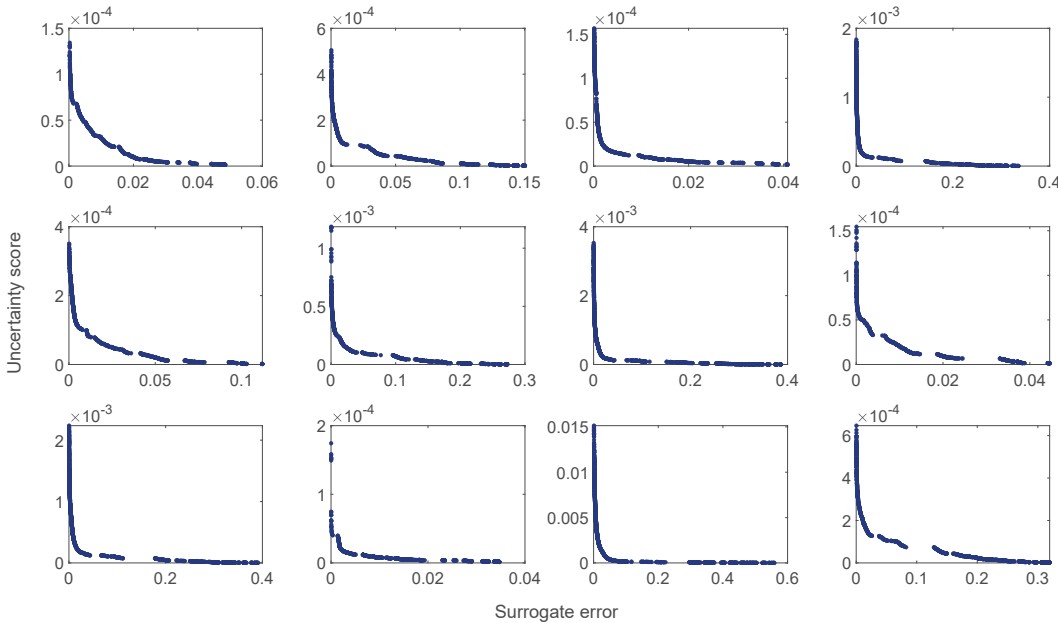

Figure 12: Pareto front for 12 randomly chosen targets.

The most interesting fact about Figures 11(a) and 11(b), is the correlation of the $\alpha$ and $\beta$ with the NFP error. This correlation means that adjusting the importance of these weights on the surrogate model directly improves the quality of the inversion in reality. Interestingly, we can observe the robustness of Autoinverse against the variation of $\alpha$ and $\beta$, such that for a range of around 3 orders of magnitude the NFP loss remains stable around a desirable value. The larger the weights, Autoinverse chooses less and less samples in the problematic regions (LC channel density larger than 0.4). This trend continues with very large uncertainty weights. However, these weights cannot be indefinitely increased as the MSE term of the objective (Equations 6 and 10 in the paper) will be undermined and inversion's NFP error increases.

## F  Pareto front of accuracy versus uncertainty

We calculate the Pareto front for 12 randomly chosen targets from the spectral printer experiment with the standard dataset (Figure 12). We use the NSGA II [8], an evolutionary algorithm that samples our forward BNN to discover the Pareto front iteratively. The uncertainty score in this experiment is the weighted sum of aleatoric and epistemic uncertainty. We set the values of the weights on the tuned values on the inversion task. The population size and the number of generations in this experiment are 1000 and 100, respectively. Figure 12 suggests that the losses of uncertainty aware inversion are conflicting such that for example reducing the MSE loss will lead to the increase of the uncertainty score.

## G  Ablation of the number of networks in the ensemble

In this experiment we investigate the importance of the accuracy of the calculated uncertainties on the final NFP error. We have trained deep ensemble networks with a varying number of networks in the ensemble. The networks are trained on the spectral printer experiment for both noisy and sparse datasets. As evident in Figure 13, by increasing the number of networks in the ensemble the NFP error improves significantly. This trend indicates the importance of accurate prediction of the uncertainties.

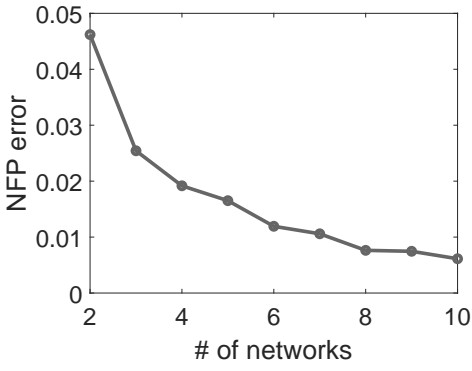
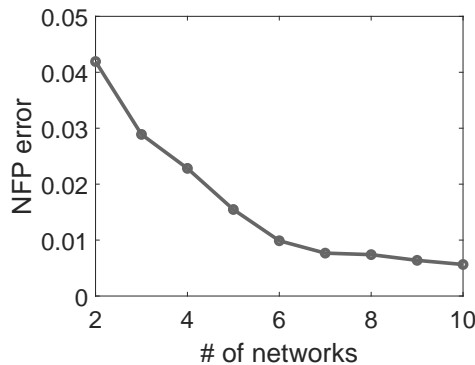

(a) The effect of increasing the number of networks in the ensemble trained on the noisy data.

(b) The effect of increasing the number of networks in the ensemble trained on the sparse data.

Figure 13: Evaluating the inversion performance using Deep Ensembles with different number of sub-networks.

## H Implementation

In practice, training the ensemble networks directly with negative log likelihood loss (Equation 8) is challenging [31]. Instead, following [29], we take a 3-step procedure for implementing deep ensemble predictive uncertainty. We start with training an ensemble of conventional networks with diverse activation functions and MSE as its loss. These networks are in fact the initialization of $\mathbb{F}_\mu(\cdot)$. The next step is training $\mathbb{F}_\sigma(\cdot)$ and fine tuning $\mathbb{F}_\mu(\cdot)$ jointly with the negative log likelihood loss (Equation 8). Finally, we replace $\hat{f}(\cdot)$ with $\mathbb{F}_\mu(\cdot)$ and incorporate $\mathbb{F}_\sigma(\cdot)$ in the `Autoinverse` loss.