# OpenReview forum: "Autoinverse: Uncertainty Aware Inversion of Neural Networks"
_NeurIPS.cc/2022/Conference — NeurIPS 2022 Accept_

### Official Review · Reviewer_uj48 · 2022-07-02

**Rating:** 7
**Confidence:** 4
**Soundness:** 3 good
**Presentation:** 3 good
**Contribution:** 3 good

**Summary:**

The paper discusses an approach for inverting surrogate functions, based on neural networks, in a face of (epistemic and aleatoric) uncertainty.

**Questions:**

1) Section 2 (Related work) would be expected to place  this work within the current state-of-the-art, however, it does not look like it fully does: how do the state-of-the-art works address the problem of estimation of uncertainty, what prior methods address this problem and what is the closest work which the authors build or improve upon? It is said that ‘In this method, a specialized architecture built upon normalizing flows [9] is trained   simultaneously in both forward and inverse directions leading to a bijective mapping between design and performance spaces. “ but it is not followed up in the following sections.  Furthermore, the similarity between the proposed ensemble architecture and normalising flows is unclear to the reviewer. (Also, on this sentence, the reviewer has probably misunderstood how exactly could the proposed modification of normalising flows be trained in both directions, if the original normalising flows [Resende &Mohamed, 2015] are invertible-by-design?)
2) The reviewer struggles to follow section 3 (Method): what is the precise model architecture, for example? (while there is the code so that the model is reproducible, it is still important to discuss and justify main aspects of implementation) While Appendix gives implementation details (Section 1 of the Appendix), it is unclear what the relation to neural adjoint method is and how does the proposed model build upon it; furthermore, it would be good if the paper references the descriptions in the appendix.
3) In Eq. 4-5, it is important to clarify, why the optimisation happens with respect to the inputs X, and where and how are the parameters of the model optimised.
4) “Following [27], we take a 3-step procedure for implementing deep ensemble predictive uncertainty. “ What would the three steps be?
5) On the experiments side, it is important to describe the experimental tasks in more detail to make it reproducible (which are not Ava; the reviewer thinks that the paper should describe what exact hyperparameters and settings correspond to the experimental evaluation.
6) It is important to identify (and compare with) the state-of-the-art methods; if it is not possible for whatever reason, there should be a convincing argument why such comparison would not be valid. I accept that Table 1-2 present evaluation against Neural Adjoint method, but just two comparisons on two tasks do not indicate the advantages (and also importantly, restrictions) of the proposed method.
7) (Not as critical as the previous points but still) The concluding remarks state: ''We look forward to see Autoinverse adopted for more inverse architectures, beyond tandem and more optimizations (beyond first-order NA). The Autoinverse cost function is multi-objective. It is highly interesting to capture the trade-off between accuracy and uncertainty using the Pareto front. '' It does not give much summary of the main conclusions of the paper, and also the future work suggestions are difficult to follow (more inverse architectures/optimisations: in which sense? Pareto front: would be interesting to see it, could be actually quite nice to see them as an additional experiment, e.g. in the appendix)


**Limitations:**

There is a need to discuss the limitations of the work more; one way to address it would be to use standard benchmarks such as the ones proposed in the standard invertible neural networks literature (e.g. MNIST and CIFAR experiments from Rezende and Mohamed, 2015, or experimental section of Grathwohl et al, 2019); otherwise, it would be useful to state why such experiments are not like-for-like or not possible.

* Rezende D, Mohamed S. Variational Inference with Normalizing Flows. In: International Conference on Machine Learning, 2015 (pp. 1530-1538). PMLR.
* Will Grathwohl*, Ricky T. Q. Chen*, Jesse Bettencourt, Ilya Sutskever, David Duvenaud. "FFJORD: Free-form Continuous Dynamics for Scalable Reversible Generative Models." International Conference on Learning Representations (2019).

**Strengths And Weaknesses:**


Strengths :

- The problem statement: quantification of uncertainty for invertible processes is an important problem
- The idea of modelling uncertainty as outlined in Section 3.1 is a reasonable one
- The authors plan to release the spectral printer dataset; the new dataset would be beneficial for the research community


Weaknesses:
- The writing looks not entirely clear (see the question section)
- Novelty could also be emphasised as it is unclear where the proposed method stand amongst the state-of-the-art models such as variational autoencoders and normalising flows (see q1 in the questions section)

I put the score as 'reject' for now, but looking forward to the rebuttal and hope the authors could address the questions appropriately.

***

The score is updated as the concerns have been thoroughly addressed during the rebuttal (see the discussion below).

---

> ### Author Response · Authors · 2022-08-02
> **Answer to Reviewer uj48**
>
> Thank you for the thorough and thoughtful review. We appreciate the raised questions that help us clarify the paper. We provide responses to the raised questions below.
>
> **Clarifying why in Equations 4-5 the optimization happens with respect to x**
>
> To recap our proposed method, Autoinverse starts with training a BNN (Deep Ensembles in our case) capable of a prediction and its uncertainty. Then we freeze all the weights and biases of the network and perform the inversion. Taking neural adjoint (NA) as the main example, its inversion loss is described in Equations 4 and 5 (original paper). UANA (Autoinverse applied on NA) is built upon NA by replacing the conventional forward model in NA with a BNN. Its inversion loss (Equation 6, original paper) then includes not only the accuracy term but also that of uncertainty.
> With this brief overview, we optimize for x in Equations 4-5 (and similarly 6) because we are looking for designs x which are the input to our forward network. Note that two important steps in Autoinverse, i.e., training the BNN and the inversion are performed sequentially and independently. In order to make this clear, we have reordered and reworded the method section.
>
> **Emphasizing the novelty of our work.**
>
> Please see the list of contributions in the Review Summary. We will stress these novelties in the revised paper.
>
> **How do state-of-the-art works address the problem of estimation of uncertainty?**
>
> To the best of our knowledge, we are the first to perform the uncertainty-aware inversion.
> We have discussed a variety of state-of-the-art methods for predicting the uncertainty as well as state-of-the-art inversion methods in the related section. To further expand our literature review, we will include more references on the potential pitfalls of the deep ensemble uncertainty estimation and the advantage of considering aleatoric and epistemic uncertainty separately.
>
>
> **The relation of the normalizing flows to our work.**
>
> Apologies for the vague wording. Normalizing flows was merely mentioned in our work as the basis of one of the state-of-the-art inversion methods (INN) and does not have any relation to Autoinverse. We have reworded (and removed part of) the misleading paragraph to avoid this confusion.
>
>
> **Three steps of implementing Autoinverse.**
>
> The first step starts with ‘we start with …’ (L135 of the original paper), the second starts with ‘The next step …’ (L137), and the third starts with ‘ Finally, …’ (L138).
> This part is reworded in the revised version to be more clear. This section is moved to SM and will be expanded.
>
> **Details of the experiments and reproducibility.**
>
> Section 4.1 of the original paper describes the details of the experiments. Moreover, in sections 2, and 3 of the original SM we described the conditions under which the experiments in the main paper are performed. Tables 1, 2, 4, 5, 6, and 7 carry the details of all the networks that we have used in these experiments. All these details are accompanied by the code to ensure the reproducibility of our experiments.
>
>
> **Future works are difficult to follow.**
>
> We have now expanded the discussion section in the revised paper to further elaborate on the future work and limitations.
>
> **Pareto front.**
>
> We have evaluated the Pareto front of accuracy vs. uncertainty score in Section 6 of the new SM. We have calculated the Pareto front for 12 randomly chosen targets from the spectral printer experiment with the standard dataset. We use the NSGA II algorithm to sample our forward BNN and improve the Pareto front iteratively.
>
>
> **Evaluation on the benchmarks used in standard invertible architectures.**
>
> There are two main differences between our proposed method and the standard invertible architecture.
> Our method is not concerned with the generative behavior in inversion. Note that in the context of inverse design for fabrication and inverse kinematics (which are the main focus of our work), we are mostly interested in finding a single solution with respect to some criteria. Our best solution in this work is the one with maximal accuracy and minimal uncertainty.
> The second difference is that we are evaluating the quality of the results with respect to the NFP error, while in the generative benchmarks, we merely assess the quality of the generated samples (surrogate error in our context).
> Instead, we performed our experiments on a variety of applications previously used as benchmark by important papers. This includes:
> - Spectral printing
> - Inverse kinematics of soft robots
> - Multi-joint robot

---

> > ### Comment · Reviewer_uj48 · 2022-08-05
> > **Really good job with rebuttal**
> >
> > Overall, the paper is in a much better shape now, addressing the comments from me and other reviewers. I'm updating the score accordingly to recommend acceptance.
> >
> > **Clarifying why in Equations 4-5 the optimization happens with respect to x:** the rewording helped understand the motivation.
> >
> > **Please see the list of contributions in the Review Summary. We will stress these novelties in the revised paper.** That is made clear now, the only question would be to somehow emphasise if there are theoretical justifications for the claim "Enforces the feasibility of solutions as infeasible solutions would contribute to high uncertainty?" Visualisation in Figure 3 helps greatly towards the explanation, however, one might say that the claim is too strong for the evidence: it gives intuitive explanation why this would happen but does not definitively show that it happens.
> >
> > **How do state-of-the-art works address the problem of estimation of uncertainty?** The comment shows the advantages of the proposed problem statement, in combination with the answer to similar concerns from ijXf and Section 3.4 of the supplementary materials, which discuss the tradeoffs between epistemic and aleatoric uncertainty.
> >
> > **The relation of the normalizing flows to our work.** The suggested amendments look good.
> >
> > **Details of the experiments and reproducibility.** The comments, amendments with the ablation studies and explanations look reasonable.
> >
> > **Evaluation on the benchmarks used in standard invertible architectures.** Following the changes in the paper structure, that follows from the amended text. Thank you for the changes in the description.

---

> > > ### Author Response · Authors · 2022-08-08
> > > **We are glad that we could address the doubts of the reviewer, and we appreciate the revision of our score.**
> > >
> > > ### Theoretical justifications for the claim "Enforces the feasibility of solutions as infeasible solutions would contribute to high uncertainty?".
> > >
> > > The justification for this statement is scattered throughout the paper and the supplementary material (we will bring them together in the revised version). Here we try to summarize it:
> > >
> > > **The epistemic uncertainty increases with the scarcity of data (reference [17] in the paper).**
> > > In the deep ensembles paradigm, as we query unknown regions (i.e., regions that are not represented in the training data), each individual network in the ensemble starts to behave differently. This divergence in behavior is projected through (Equation 10c).
> > > Equation 10c is the same term that appears in the objective of the inversion (Equations 3 and 7) and stops it from choosing the designs in regions with high epistemic uncertainty.
> > >
> > > To put it concretely:
> > > 1. The epistemic uncertainty improves with more data (reference [17] in the paper).
> > > 2. By definition, we do not have any infeasible samples in the training data (paper: L255-256, SM: L82-83).
> > > 3. As a result, the epistemic uncertainty is higher in out-of-range regions (paper: L238, L254-256, SM: L79-81, Fig4-A).

---

### Official Review · Reviewer_xeN4 · 2022-07-10

**Rating:** 6
**Confidence:** 4
**Soundness:** 3 good
**Presentation:** 3 good
**Contribution:** 2 fair

**Summary:**

The paper presents an approach to solving inverse problems, i.e. find parameters $\mathbf x$ that explain/produce given observations $y$ when passed through the forward model $f(\mathbf x)$. The approach explicitly models and accounts for the uncertainty of the forward process by learning a surrogate of it using a deep ensemble. The deep ensemble models the forward process as $y \sim N(\mathbb F_\mu(\mathbf x), \mathbb F_\sigma(\mathbf x)$, with $\mathbb F_\sigma(\mathbf x)$ explicitly modelling the uncertainty of the forward process. The   $\mathbb F_\sigma(\mathbf x)$ is then plugged within two inversion methods, the neural adjoint and a tandem architecture, producing their uncertainty-aware variants. As a result when looking for the inversion solution the method will avoid $\mathbf x$ which come with a high uncertainty with the $y$ that they produce. The two methods are evaluated in three benchmarks and demonstrate good inversion performance.

**Questions:**

In relation to the single-point vs the complete posterior distribution it would be useful to discuss how the single point solution fits within the complete posterior distribution, can we make any other statement on where the point estimate x relates with the complete p(x|y) other than that it is an $x$ for which $p(y|x)$ has the least uncertainty?

In the evaluation measures I am not sure I get correctly what the surrogate error is and what the NFP error is. Is the surrogate error $|| f(f^{-1}(y)) - y||$ or is it $||f(\mathbf x) - y||$, in other words does it measure the gap between the prediction of the forward surrogate in the inverse solution, or the gap on paired training instances (x,y). In what concerns the NFP error, is this the $|| NFP(f^{-1}(y)) - y ||$, i.e. the error we get when we pass through the real forward process at the inverse solution?

With respect to the INN wouldn't a more appropriate way to get the inverse solution be the MAP estimated of the p(x|y) instead of getting empirical samples from it?

**Limitations:**

The authors adequately addressed the limitations and potential negative societal impact of their work.

**Strengths And Weaknesses:**

The paper is rather well written. The main idea of the paper is that when looking for inverse solutions, $\mathbf x$, that could produce a given observation $y$ one should avoid $\mathbf x$ that come with high uncertainty for their predicted $y$, as quantified by the deep ensemble and the estimated $\mathbb F_\sigma(\mathbf x)$ parameter. In fact $\mathbb F_\sigma(\mathbf x)$ can be decomposed in two terms one measuring aleatoric uncertainty and the other the epistemic uncertainty. The latter is the extend to which the components of the ensemble differ with respect to the average prediction, i.e. the $\mathbb F_\mu(\mathbf x)$. A high epistemic uncertainty can be evidence of the $\mathbf x$ strays away from the training distribution and thus the base models produce very different predictions. Incorporating the term in the loss functions keeps the search for an inverse solution away from such high uncertainty regions in which the learned surrogate is probably not a very accurate proxy of the true forward process.

If I understood correctly the paper only provides a point estimate, i.e. a single $x$ given a desired $y$, and along the above discussion the one with the least uncertainty. To my understanding in solving inverse problems one is also strongly interested in accessing the full posterior of $\mathbb x$, i.e. $p(\mathbb x|y)$, something that is delivered by generative methods such as the invertible neural networks which the authors cite and compare as one of the baselines. There is no discussion, unless I missed it, on the quite point estimate vs posterior distribution approach; this is a weakness of the paper, and one could also say a weakness of the method.

---

> ### Author Response · Authors · 2022-08-02
> **Answer to Reviewer xeN4**
>
> Thank you for the thorough and thoughtful review. We appreciate the detailed feedback. We provide responses to the raised questions below.
>
> **Discussion on point estimate vs. posterior distribution.**
>
> We agree that, in many tasks, finding the distribution of solutions is preferred. However, in the context of inverse design for fabrication, and inverse kinematics -which are the main focus of our work- we are mostly interested in finding a single solution with respect to some criteria: maximal accuracy and minimal uncertainty. Currently, we balance the importance of these two goals using uncertainty weights. Nevertheless, we can still give a 'generative' power to our method by:
> - Not using weights inside our optimization and solving the original multi-objective problem to find a set of trade-off solutions, known as Pareto set. This is done in Section 6 of the revised supplementary materials (SM).
> - Solving our non-convex problem multiple times using various initializations (see next question).
>
> **Relation of the single point solution within the complete posterior distribution.**
>
> To answer this question, we solved the inverse optimization 2000 times with different initializations to approximate the posterior of the target samples in the spectral printer experiment. We then show the solution with minimum uncertainty within this distribution. We could not find any particular pattern for these points within the posterior. You can find the results in a zip file uploaded alongside the SM. We have selected random samples for noisy, sparse, and standard data. In each folder, you can find the distribution for 8 channels of the printer, and the best solution is identified with a red line.
>
> **Clarification of the NFP and surrogate error.**
>
> The reviewer correctly describes the NFP error. Your first guess is also the correct description of the surrogate error: the difference between the 're-prediction' of an obtained inverse solution by the neural surrogate and comparing it with the target. We clarified our definition of surrogate error (L195 of the revised paper).
>
> **MAP estimate instead of the empirical samples.**
>
> The main evaluation criterion in Table 1 is the accuracy of each model with respect to the NFP and surrogate error. To err on the side of caution, we report the best possible error that any sample within the INN's posterior produces.

---

### Official Review · Reviewer_ijXf · 2022-07-11

**Rating:** 7
**Confidence:** 4
**Soundness:** 3 good
**Presentation:** 3 good
**Contribution:** 2 fair

**Summary:**

(Updated score post rebuttal)

The paper proposes a new method AutoInverse in the field of inverting neural networks that incorporates the uncertainty in the modelling of the NFP (natural forward process) though a neural network surrogate. By adding the epistemic uncertainty of the surrogate function (due to mismodelling) and aleatoric uncertainty in the training data (noise in the data), the new method favours points that have low uncertainty, and therefore are a better match to the NFP. Incorporating uncertainty in the inversion loss also results in desirable behaviour, such as 1) enforcing the feasibility in solutions, as points with low feasibility should have high uncertainty, 2) not requiring regularisation techniques, and 3) not requiring careful initialisation.

**Questions:**

1. I feel like the authors need to incoporate a few more papers in their literature survey regarding the modelling of aleatoric and epistemic uncertainty using a heteroskedastic regression loss. The decomposition shown in the paper has been shown before, in [1]. There has been work also showing that using this formulation can have pathological behaviour [2].
2. I found this paper lacking an adequate explanation of the tradeoff between aleatoric and epistemic uncertainty terms. The authors mention that "We observe that with a larger epistemic uncertainty we obtain better results", but I think it's important to have a better ablation here. An easy way of "improving" epistemic uncertainty estimates is to use more deep ensembles, and see how increasing the number of ensembles affects the choice of $\alpha$ and $\beta$ parameters. Similarly, by adding more noisy data, in the "Locally noisy data set" ablation, one can measure the importance of the aleatoric term. I think having these two sweeps in the ablation, along with $\alpha$ and $\beta$, will really help understand the differences between the two terms.
3. Given that the main contribution of this paper is a way of incorporating uncertainty estimates into the inversion cost function, I think in order for the paper to be a good candidate for acceptance, there needs to be a better ablation for different uncertainty estimation methods. I've mentioned more of this in the Significance section.

Minor nits:
- Line 75: "neural"
- Line 112: "push"
- Line 313: "similar"

**Limitations:**

I believe the authors should address the additional compute requirements needed for their method. Deep ensembles are the least efficient way of obtaining uncertainty estimates, and can result in significantly more GPU hours for training and inference. These should be addressed.

**Strengths And Weaknesses:**

1. Originality

The Autoinverse technique has two components: 1) Replacing the surrogate neural network function with a neural network function that is capable of predictive uncertainty, and 2) modifying the loss function used to find accurate designs given a pretrained surrogate function with frozen parameters.

The first component of this technique is a well-established baseline in literature, and deep ensembles have been very well studied in their ability to accurately model predictive uncertainty in a bunch of different domains and tasks. In fact, the decomposition of the uncertainty learnt using deep ensembles (or other methods that model heteroskedastic regression) into components that model aleatoric and epistemic uncertainty are also well-studied [1]. However, the authors do not cite [1], or discuss any potential pitfalls of modeling noise in such a way (see [2] as an example.)

The second component is the novel contribution for this paper, in my uncerstanding. By using a neural network that is modeling both heteroskedastic aleatoric uncertainty and epistemic uncertainty, this method can use these estimates as auxilliary terms in the inverse cost function, to directly influence the training of the designs. By adding epistemic uncertainty to the inverse cost function, points that have high epistemic uncertainty are downweighted (in a sense), and these are usually points where the surrogate is not performing well or is performing pathologically. By adding aleatoric uncertainty to the cost function, points with high data noise are also downweighted. This simple addition of two terms to the cost function results in quite impressive empirical performance.

2. Quality

In my opinion, the paper is well-written, has good motivations for the problem it is tackling, and has strong ablations that answered quite a few of my initial doubts/hesitations about the empirical strengths of the technique.

I have some concerns regarding the thoroughness of uncertainty quantification methods studied, and a careful analysis of the strengths and weaknesses of different methods. For example, Deep Ensembles are known to be state-of-the-art in uncertainty quantification, but have much higher compute time and storage requirements than other methods. The authors mention that the longer training time for ensembles is not a concern, as they are trained parallely. I disagree with this point, as I feel like acknowledging the GPU hours required to train a method are quite important, especially with big compute quite easily getting out of hand to obtain SOTA. Even though there are certain groups with the resources to trivially parallelise ensembles, if Autoinverse is to be established as a strong contender, the added compute time, memory footprint, and inference time needs to be acknowledged.

Another factor here is maybe exploring a few other uncertainty quantification techniques as an ablation in the paper (MC Dropout, more efficient ensembles etc.). I feel like this comparison is important to understand the contribution of adding an aleatoric and epistemic uncertainty term to the cost function, vs the quality of these uncertainty estimates w.r.t the compute required. It might be possible that deep ensembles are overkill for the inversion problem, and a lot of compute is used to obtain uncertainty estimates that might be approximated more cheaply. As the authors mention, it is a very interesting idea to capture the Pareto front between accuracy and uncertainty in an inversion task.

3. Clarity

The paper is quite easy to follow, and the experiments chosen make logical sense. I would argue that it is more important to describe the inversion technique and cost function before talking about uncertainty quantification, as it is maybe more prudent to motivate the exact problem that the authors wish to tackle. In the current ordering, it might seem that the authors use a well-known methodology of uncertainty quantification, and find a usecase that works well. Though this is obviously not true, a reordering of the sections might help mitigate this.

4. Significance

Uncertainty quantification in deep neural network applications is of fundamental importance. In order for DNN methods to be applicable in safety-critical domains, it is very important that we measure their performance, calibration and out-of-distribution robustness. To my knowledge, there has not been prior work that adequately addresses the surrogate-NFP mismatch when formulating the cost function for inversion. This paper's application is quite significant, and I believe that the empirical results show that modelling uncertainty in a principled way can result in multiple benefits. However, I feel like this paper is lacking in terms of adequately populating the Pareto front between uncertainty and accuracy, by proposing a single data point (ensembles). While the current proposed method cannot be argued with in terms of empirical performance, I feel like this paper might be a good place to populate more points here, to establish a better understanding for the community, on how the quality of uncertainty estimates affect the performance of inversion.

[1] Kendall, Alex, and Yarin Gal. "What uncertainties do we need in bayesian deep learning for computer vision?." _Advances in neural information processing systems_ 30 (2017).

[2] Seitzer, Maximilian, et al. "On the pitfalls of heteroscedastic uncertainty estimation with probabilistic neural networks." _arXiv preprint arXiv:2203.09168_ (2022).

---

> ### Author Response · Authors · 2022-08-02
> **Answer to Reviewer ijxf**
>
> Thank you for the thorough and thoughtful review. We are heartened to know that you believe our method is novel and has important applications. We provide responses to the raised questions below.
>
> **Populating the Pareto front between uncertainty and accuracy.**
>
> Following the reviewer's suggestion we have computed the Pareto front of accuracy vs. uncertainty in Section 6 of the revised supplementary materials (SM). We have calculated the Pareto front for 12 randomly chosen targets from the spectral printer experiment with the standard dataset. We use the NSGA II, an evolutionary algorithm that samples our forward BNN to discover the Pareto front iteratively.
>
>
>
> **Additional References**
>
> We initially faced the same instabilities as in [2] (e.g., bad fitting of $\mu$ and imprecise $\sigma$ values). However, the 3-step training procedure (original paper, Section 3.4) mitigated the issues to a good degree and resulted in a more stable training. We agree on the importance of addressing these pitfalls and their potential remedies. We have included [2] in the implementation section and will discuss it in the related work. We now properly refer to [1] when discussing the separation of aleatoric and epistemic uncertainties.
>
> **Ablation on $\alpha$ and $\beta$.**
>
> We had included an ablation of uncertainty weights $\alpha$ and $\beta$ in Section 5 and Figure 8 of the original SM. Given the main focus of our work is to achieve minimal NFP error, we were curious to see how too small or too large values of the weights affect the NFP error. For  $\alpha$ (aleatoric uncertainty weight), we performed inversion on the network trained with the noisy spectral printer dataset. Likewise, for the ablation of $\beta$ we performed inversion on the network trained with the sparse spectral printer dataset. In both ablations, we fixed the other weight on its tuned value.
>
> Figure 8 (of the original SM) suggests that too small weights for both noisy and sparse experiments lead to samples from unreliable regions of the network, which translates to large NFP error.
> On the other hand, too large weights disturb the objective's balance and undermine other optimization terms (e.g., MSE loss).
>
>
>
> **Do we need so much training resources?**
>
> As suggested, we have included an ablation study for the effect of the number of networks in the ensemble. We gradually reduced the size of the ensemble from 10 to 2  and performed inversion for each step. We run this experiment for both noisy and sparse datasets. As we can see in Figure 9 of the newly submitted SM, the NFP error strongly correlates with the ensemble size, which indicates the importance of accurate uncertainty prediction.
>
> Furthermore, we repeated our tuning procedure for each new ensemble but did not notice any significant alteration in the tuned $\alpha$ and $\beta$ values. In other words, if we keep the tuned values for the inversion problem using the 10-member ensemble for the rest of the experiment, we do not lose much.
>
> We hope these results resolve your doubts; otherwise, please elaborate more on the other ways the $\alpha$ and $\beta$ variations are interesting for you.
>
> **Processing capacity and the GPU time.**
>
> In Tables 2, 5, and 7 in the original SM, we have thoroughly reported the training and the inversion times for all the methods and experiments. To make it fair, we reported these times assuming serial execution of the operations (L 41-46 original SM).
> Nonetheless, we have elaborated more on this limitation in the paper (see Discussions section in the revised paper).
>
> **Monte Carlo dropout as an alternative to Deep Ensembles (DE).**
>
> Given that the focus of our work is the neural inversion, we shaped the paper around this idea and went with a safe choice (high accuracy, ease of implementation) for the forward BNN. As correctly mentioned, however, DE requires more processing capacity that might not be available to everyone. Indeed, Monte Carlo dropout could be a potential low-cost alternative. We have started the implementation of uncertainty-aware inversion using Monte Carlo dropout, and the preliminary results look promising. We start by training a forward network (similar to the inset figure in section 3.3) that contains diverse activation functions (following: L 313-321 originally submitted paper). The procedure of training the network with the NLL loss remains similar to Section 8 of the revised SM. We then extract 10 sub-networks for $\sigma$ and $\mu$ from the main networks, each of which has undergone dropout and has a different combination of nodes. These sub-networks comprise our DE, and the rest of the inversion remains similar to the UANA method. We look forward to including Monte Carlo drop-out inversion in the SM alongside some evaluations.

---

> > ### Comment · Reviewer_ijXf · 2022-08-09
> > **Response to authors**
> >
> > I thank the authors for their comprehensive rebuttal. I believe a lot of my concerns were adequately addressed, and I thank them for running experiments that I understand are difficult to get running in a short time. I think these results make the paper stronger, and I am updating my score accordingly!

---

### Author Response · Authors · 2022-08-02
**Revision summary**

We thank the reviewers for their thoughtful feedback. We are delighted to hear that our work has “impressive empirical performance” and " significant applications", includes "strong ablations", and is “easy to follow” and “well-written”. We summarize here the main aspects of our rebuttal followed by our detailed response to each reviewer.

To reiterate our contributions: our paper advocates a new philosophy in which the gap between an original process (NFP) and its data-driven surrogate could be closed in a highly automated manner by accounting for uncertainty during the inversion process. In addition to obvious accuracy gains, it has deeply important implications. Autoinverse:
1. Enforces the feasibility of solutions as infeasible solutions would contribute to high uncertainty,
2. Can be run with arbitrary initializations,
3. Without any explicit regularisation, performs on par with methods that have explicit regularizations, hand-crafted by human experts.

Although our paper builds on existing BNN forward models, its main message lies in its inversion strategy. To the best of our knowledge, and as evident from the reviews, both the strategy and its implications don't have any precedent in the literature.

Finally, we believe the 'scalability' of our proposed methodology is one of its important advantages. Future research can:
1. Progress in making more optimization- and architecture-based inverse methods uncertainty-aware,
2. Study the impact of different types of BNNs on the inversion performance,
3. Go beyond neural inversion toward more general uncertainty-aware data-driven surrogate modelling.

Given the valuable feedback by the reviewers, we have made specific changes and ran additional experiments:
1. We have reordered the method section to better reflect our main contribution (a neural inversion strategy).
2. We have shown that a set of trade-off solutions (between accuracy and uncertainty) can be obtained via Pareto front computation (revised SM, Section 6).
3. We have shown that our method could be repeatedly run to generate a distribution of solutions and how the most certain solutions lie within this distribution (a large set of graphs uploaded alongside the supplementary materials)
4. We have shown the importance of allocating resources to the forward BNN by running experiments on Deep Ensembles with different numbers of sub-networks (revised SM, Section 7).
5. We are experimenting to have Autoinverse run on a totally new BNN, i.e., Monte Carlo dropout that requires less computational resources.

We have uploaded a revised version of the paper and the supplementary materials (SM). They will be further extended and polished for the final version.

---

### Meta-Review · Area_Chair_nbad · 2022-08-27

**Recommendation:** Accept
**Confidence:** Certain

**Metareview:**

Thanks to the authors for submitting this super interesting work. The reviewer discussion reflected an overall satisfaction with the submission, author responses, and updated manuscript.  The additional experiments directly addressed reviewer concerns, and contributed to increased scores and clarifications in the submission.  Given the clear consensus and reviewer enthusiasm, I recommend acceptance.  Well done!

**Award:**

No

---

### Decision · Program_Chairs · 2022-09-14

Accept